# Wood-inspired metamaterial catalyst for robust and high-throughput water purification

Lei Zhang[1,2,3,4,8], Hanwen Liu[1,8], Bo Song [1] ✉, Jialun Gu[2,3,4], Lanxi Li[5], Wenhui Shi[1], Gan Li[2,3,4,6], Shiyu Zhong[2,3,4], Hui Liu[2,3,4], Xiaobo Wang[1], Junxiang Fan[1], Zhi Zhang[1], Pengfei Wang[7], Yonggang Yao [1] ✉, Yusheng Shi[1] & Jian Lu [2,3,4] ✉

Continuous industrialization and other human activities have led to severe water quality deterioration by harmful pollutants. Achieving robust and high-throughput water purification is challenging due to the coupling between mechanical strength, mass transportation and catalytic efficiency. Here, a structure-function integrated system is developed by Douglas fir wood-inspired metamaterial catalysts featuring overlapping microlattices with bimodal pores to decouple the mechanical, transport and catalytic performances. The metamaterial catalyst is prepared by metal 3D printing (316 L stainless steel, mainly Fe) and electrochemically decorated with Co to further boost catalytic functionality. Combining the flexibility of 3D printing and theoretical simulation, the metamaterial catalyst demonstrates a wide range of mechanical-transport-catalysis capabilities while a 70% overlap rate has 3X more strength and surface area per unit volume, and 4X normalized reaction kinetics than those of traditional microlattices. This work demonstrates the rational and harmonious integration of structural and functional design in robust and high throughput water purification, and can inspire the development of various flow catalysts, flow batteries, and functional 3D-printed materials.

Continuous industrialization and other human activities have led to severe water quality deterioration by harmful pollutants, such as dyes, heavy metals, and antibiotics[1,2]. The present water purification system is multi-process and time-consuming, which includes physical processes and biochemical processes such as filtration, agglomeration, and desalination. However, physical filtrations and chemical reactions often belong to two different stages of the water purification process. The former filters out large particle residues[3,4], robustly provides the indispensable mechanical structure, and guides high-throughput water flow, while the catalyst can harmless pollutants and improves

[1]State Key Laboratory of Material Processing and Die & Mould Technology, School of Materials Science and Engineering, Huazhong University of Science and Technology, Wuhan 430074, China. [2]CityU-Shenzhen Futian Research Institute, Shenzhen 518045, China. [3]Centre for Advanced Structural Materials, City University of Hong Kong Shenzhen Research Institute, Greater Bay Joint Division, Shenyang National Laboratory for Materials Science, Shenzhen 518057, China. [4]Department of Mechanical Engineering, City University of Hong Kong, Tat Chee Avenue, Kowloon, Hong Kong, China. [5]Department of Materials Science and Engineering, City University of Hong Kong, Tat Chee Avenue, Kowloon, Hong Kong, China. [6]Shenzhen Key Laboratory for Additive Manufacturing of High-performance Materials, Department of Mechanical and Energy Engineering, Southern University of Science and Technology, Shenzhen 518055, China. [7]Advanced Materials and Energy Center, China Academy of Aerospace Science and Innovation, Beijing 100176, China. [8]These authors contributed equally: Lei Zhang, Hanwen Liu. ✉e-mail: bosong@hust.edu.cn; yaoyg@hust.edu.cn; jianlu@cityu.edu.hk

the reaction efficiency or selectivity toward target pollutants[5–7]. However, current works have separately focused on wastewater catalyst development and structural framework design[8]. In addition, due to the coupling between mechanics, transportation, and catalysis, it is difficult to achieve multi-performance collaborative improvement[9]. It is urgent and eager to develop a harmonious integrated design of functional catalysts and mechanical frameworks to achieve system-level synergistic improvements.

Metamaterials are artificial structures that can be flexibly designed to achieve special physical properties from the microscale to the macroscale[10–14]. The geometrical foundation of microlattice metamaterials originates from the study of atom lattices[15–18], which are periodically arrayed by interconnected units with connecting struts and custom pores. In the microlattice metamaterials, the struts determine the mechanical strength, while the pore size distribution influences fluid/gas transport[3,8,19–22]. Therefore, they are widely used in mechanical engineering and biology/chemistry/environment fields requiring robust and high-throughput transport regulation. Microlattice metamaterials have been applied in the design of artificial scaffolds to mimic the stiffness and the transportation of the human bone to simultaneously support human movement, nutrient transport, and metabolism, respectively[23–25]. Moreover, the rational pore distribution design of microlattices could tune the thermal transport and allow them to be used as efficient heat insulation devices[26–28]. The controllability of the multi-physical properties (e.g., mechanical and transport properties) of microlattice metamaterials allows for functional integration, flexible design, and property tunability. Thus, a water purification system with mechanical robustness and high flow throughput can be rationally achieved. However, the geometrical characteristics of the traditional periodic microlattices are highly coupled and mutually constrained, which limits the tunability of their physical properties[9,29]. High mechanical strength often corresponds to less pore distribution and, therefore, limited transport, thereby suppressing the possible design and tunable range.

Today, bionics allow microlattice metamaterials to achieve superior physical properties by mimicking natural shape, performance, and function[29–33]. For example, introducing bamboo-like hollow strut elements into a microlattice can largely enhance specific stiffness and specific strength, to approach the Hashin–Shtrikman bound[15]; designing a porous lotus root-like microlattice can provide excellent bone regeneration and repair ability to meet the strength and transportation requirements of implants[32]. Herein, we propose an innovative metamaterial design inspired by the Douglas fir. The Douglas fir can grow up to 328 feet (100 m) high but with a much smaller diameter (-1.5 m) (Fig. 1a). Such an ultrahigh but thin tree requires considerable strength to resist wind and a mechanism that allows for the sufficient transmission of water and nutrients from the root to the uppermost tip. Microstructural analysis has revealed that the key factor supporting the robust growth of the tree lies in the staggered/bimodal pore distribution pattern originating from vessels and fibers (Fig. 1b–d and Supplementary Fig. 1). The chessboard pores facilitate the use of limited volume space for matter transportation, while the staggered modes resemble sandwich structure to improve the strength, thus decoupling the mechanical-transport properties and achieving synergistic improvements and a superior combination of mechanical-transport properties, which largely differs from the regular uniform pores in traditional periodic microlattices (Fig. 1e).

Inspired by the bimodal pore size distribution of the Douglas fir tree, we used a body-center cubic (BCC) microlattice overlap strategy to construct bimodal pores (Fig. 1f). The overlap microlattice configuration features a staggered distribution of large and small pore areas (Fig. 1g). By superposing microlattices and changing the spatial shape of the internal pore region, this wood-inspired overlapping design strategy can substantially increase the degree of freedom (DOF) of

metamaterial design and the tunability of the mechanical and transport properties.

Thanks to the development of additive manufacturing technology, the manufacturing of highly complex microlattice metamaterials can be realized. However, there are significant differences between different additive manufacturing technologies in terms of material, strength, price, structure, and mechanism. Taking metal additive manufacturing as an example, the lower end of the cost spectrum, such as wire-arc additive manufacturing (WAAM), often has lower performance and accuracy[34,35]; Medium performance and moderate cost but with good accuracy, such as the selective laser melting (SLM) process using materials such as stainless steel (SS), but requiring non-conventional particle control settings and limited to manufacturing sizes[36,37]; Some of the highest-performance metal microlattice production technologies based on two-photon lithography (TPL) and electroplating processes have been fully studied[38,39], but are often highly-specialized, time-consuming and cost-intensive. In addition to the 3D printing process, the time required for processes such as polymerization, curing, electroplating, grinding, and etching may exceed 24 hours[40–42]. To meet the comprehensive requirements of the sewage treatment system for the size, accuracy, strength, transport and catalyst adhesion ability of the support frame, this work adopts a compromise SLM-based 3D printing technology for the manufacturing of 316 L stainless steel microlattice metamaterials with different strut diameters and overlap rates.

After the 3D printing process, we decorated the surface of the Ferrum (Fe)-based metamaterial with cobalt (Co) via an electrochemical deposition process to form a highly efficient sewage treatment system (Fig. 1h), which integrates efficient Co/SS catalysts and wood-inspired structural advantages (optimized robustness and high-throughput flow) (Fig. 1i). Therefore, through the architectural design and further structural functionalization, such a groundbreaking structure-function integrated interdisciplinary field is coined as a "metamaterial catalyst", combining physical-chemical properties derived from the conceptual breakthroughs of emerging metamaterials. The Co/SS-based metamaterial catalyst possesses superior capability and extended freedom in mechanical-transport-catalytic property tunability. The metamaterial catalysis endows the structure-functional integration of mechanical and transport performances, as well as the high-efficiency synergistic catalytic performance of the materials for water purification applications (Fig. 1j).

## Results and discussion
### Morphological characteristics and mechanical robustness
To comprehensively demonstrate the high strength and good transport ability of the wood-inspired metamaterial catalyst for water purification, we constructed a series of microlattice metamaterials with different strut diameters (0.30, 0.35, and 0.40 mm) and overlap rates (0, 30, 50, and 70%). 316 L SS, which is known to provide sufficient mechanical strength and corrosion resistance in water purification, was selected as the parent material. The experimental samples were prepared via SLM, which was characterized by high precision and high efficiency in the production of complex metallic structures[43–48]. SS-based microlattice metamaterials with different geometrical characteristics were successfully printed (Fig. 2a). The optical images of the 3D-printed microlattice metamaterials with 0.40 mm diameter and 0%, 30%, 50%, and 70% overlap rates are shown in Fig. 2b. Based on the Gibson-Ashby model of $E/E_s = C(RD)^n$ ($E$ and $E_s$ are Young's modulus of microlattices and parent materials, respectively, $C$ is within the ranges of 0.1–4.0, and $RD$ is the relative density, $n$ is 2 for bending-dominated structures)[49–51], there is a positive correlation between the relative density and Young's modulus of microlattice structures. The overlap design strategy was intended to increase relative density resulting from the increased node numbers and internal constraints of struts

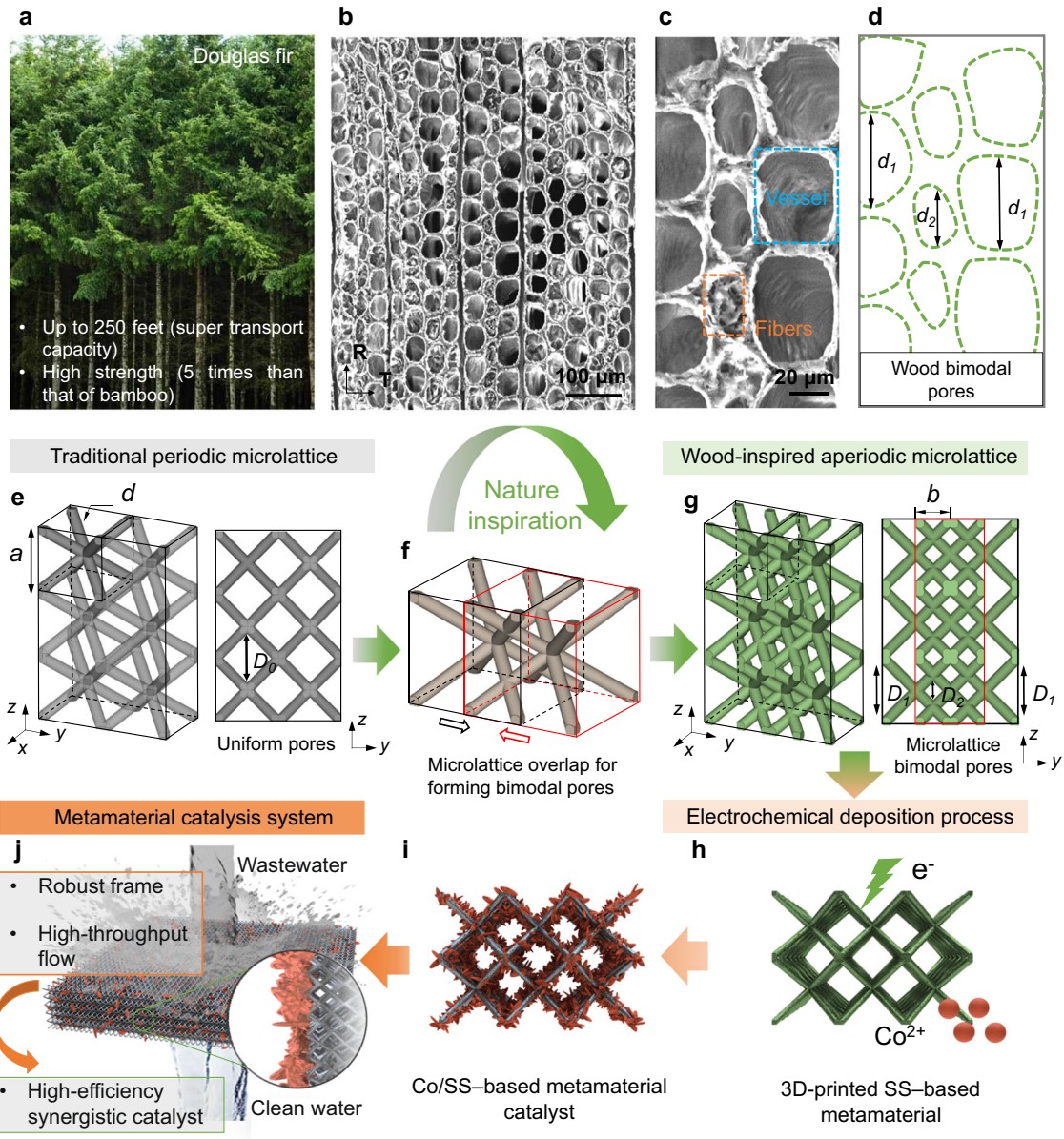

**Fig. 1 | Wood-inspired metamaterial catalyst. a** Natural Douglas fir. **b** Cross-sectional scanning electron microscopy (SEM) morphology of Douglas fir (R, radial direction; T, tangential direction). **c** Enlarged SEM images showing the staggered arrangement of different apertures. **d** Schematic of the staggered pore distribution pattern with large pores, $d_1$, and small pores, $d_2$. **e** 3D and front view of the traditional periodic microlattice with uniform pores ($D_O$). **f** Schematic of microlattice overlap process for forming bimodal pores inspired by the wood's microstructure. **g** 3D and front view of the wood-inspired aperiodic microlattice. Overlapping units feature interlaced bimodal pores (large pore, $D_1$, small pore, $D_2$), similar to the staggered pores of Douglas fir. **h** Schematic of electrochemical deposition process for the 3D-printed SS-based metamaterial in Co ion solution. **i** Synthetic Co/SS-based metamaterial catalyst after 3D printing and electrochemical deposition process. **j** Schematic of the wood-inspired metamaterial catalysis system with superior mechanical robustness, high-throughput flow, and high-efficiency catalysis for water purification.

(Fig. 2c), which could significantly improve the structural stiffness to effectively withstand the flow impact. Meanwhile, the wood-inspired bimodal pores were intended to retard the fluid velocity of wastewater, thereby increasing the liquid-solid contact time in the matter transportation process for high-efficient catalytic purification. This is due to the wood-inspired overlap design increasing the surface area per volume, thereby increasing flow friction (Fig. 2c).

As shown in Fig. 2d, e, the traditional microlattice features uniform square pores with thin walls that were made up of inclined struts and robust nodes. After the overlap design of wood-inspired bimodal pores, the shaped pores sprout from staggered microlattice cells. Figure 2f, g show that numerous small pores are generated between neighboring microlattice cell walls (in the middle region) due to the overlap of space volume, resulting in decreased structural porosity. The comparison of the micro-CT-reconstructed and CAD models of traditional microlattices and wood-inspired metamaterials reveals that a surface deviation occurred on the downward surface of the struts, as indicated by the red nephogram, owing to the lack of sufficient holding force (Supplementary Figs. 2–3). Even if there are manufacturing deviations in 3D printing technology, the overall structure remains highly accurate with a surface peak deviation ($D_p$) of less than 30 μm and a dense solid part with small void defects (Fig. 2h, Supplementary Figs. 4, 5, and Supplementary Note 1). We have carefully considered the potential impact of manufacturing defects, such as void defects,

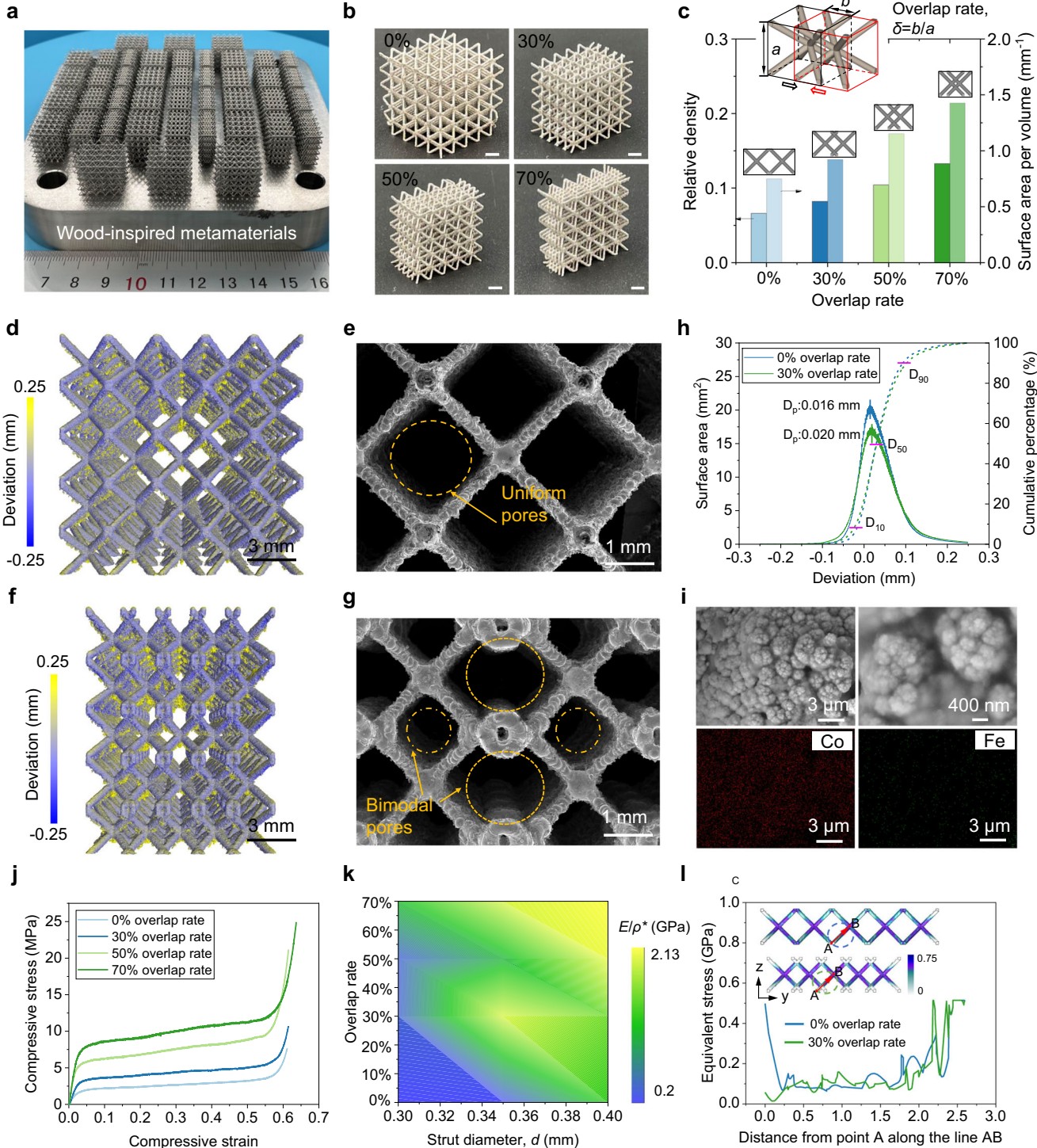

**Fig. 2 | Morphological structure and mechanical properties of the metal 3D-printed traditional microlattices and wood-inspired metamaterials. a** Optical image of original 3D-printed traditional microlattices and wood-inspired metamaterials. **b** The 3D-printed metamaterials with 0.40 mm diameter and 0, 30, 50, and 70% overlap rates. **c** The relative density and surface area per volume of traditional microlattices and wood-inspired metamaterials. The insert schematic shows the definition of overlap rate. Comparison of the micro-CT-reconstructed and CAD models of **d** traditional microlattices and **f** wood-inspired metamaterials. The SEM morphology of **e** traditional microlattices and **g** wood-inspired metamaterials. **h** Distributions of surface deviations. The D-values ($D_{10}$, $D_{50}$, and $D_{90}$) indicate the intercepts for 10, 50, and 90% of the cumulative percentage. **i** The microstructure

of Co on the 3D-printed 316 L microlattices. **j** Stress–strain curves of 3D-printed traditional microlattices and wood-inspired metamaterials with different overlap rates under compression perpendicular to the overlap direction. **k** Contour map of normalized Young's modulus under different overlap rates and strut diameters for the wood-inspired metamaterial. **l** The localized equivalent stress distribution comparison between traditional microlattice and wood-inspired metamaterial with a 30% overlap rate. The material properties of SLM-fabricated 316 L SS for the compression simulation are shown in Supplementary Fig. 13. The top plate only was allowed to move along the $z$-axis while the bottom plate was fixed in all directions. Fig. 2c, h, j–i are provided as a Source Data file.

cracks, and manufacturing accuracy, in the SLM process on the performance of metamaterial catalysts and discussed their influences in Supplementary Figs. 5–10 and Supplementary Notes 2–5. Given the maturity of SLM technology and the high additively manufacturability of 316 L SS metal powder, the few defects in both SLM-printed microlattices and wood-inspired metamaterials can be ignored. In addition, the small unevenness of the 3D-printed metamaterial surface facilitates the increase of surface area and the adhesion of Co atoms of the electrochemical deposition process. It can be found that there is a large amount of Co on the surface of the 3D-printed 316 L metamaterial surface (Fig. 2i and Supplementary Fig. 11).

The application of wood-inspired microlattice metamaterial catalysts requires evaluating the load-bearing capacity and its ability to withstand the impact of water flow. Damage-resistant and high-strength metamaterial catalysts are conducive to the long-term usability and stability of water purification systems. To explore whether the microlattice overlap process changes did impact the mechanical robustness, we used a quasi-static compression experiment to measure Young's modulus of the wood-inspired metamaterial catalyst under compression (Fig. 2j, k). All of the 3D-printed microlattice metamaterial catalysts exhibited a stable plateau stage after the linear elastic stage of rapid stress increase under longitudinal external loads, and then the final densification stage was accompanied by a sharp rise in stress (Fig. 2j). We found that the larger overlap rate of microlattices, the higher strength at a same strain. The Young's modulus of traditional microlattices of 0.40 mm strut diameter with a 0% overlap rate is 110.17 MPa, which sharply increased to 367.36 MPa as the overlap rate increased to 70%. In addition, the variation of strut parameters in the microlattice metamaterials does not change the robustness-enhanced attribution of the microlattice overlap (Fig. 2k and Supplementary Fig. 12). Due to the flexibility of the BCC-type microlattice and the plasticity of the 316 L material, metamaterial structures all failed as the buckling deformation Supplementary Figs. 13, 14). The finite element visualization results reveal that the stress was mainly concentrated near the node (Fig. 2l). The mechanical analysis of Euler's buckling theory also shows that the decrease in equivalent strut length results in an increase in the buckling force, therefore leading to increased mechanical robustness (Supplementary Fig. 15). The proposed microlattice overlap design strategy could effectively disperse the stress under longitudinal external loading to improve the overall structural strength. With increasing overlap rate, the constraint between nodes in the wood-inspired microlattice metamaterials increased and the structural stiffness increased. Our results show that the overlap process and subsequently forming the bimodal pores greatly improves the robustness of microlattice metamaterials. At the same time, we should note that for different topological microlattices, the influence of this wood-inspired overlapping design strategy on their mechanical properties is distinctive. In addition to the BCC-type microlattice, we also considered the mechanical responses of other topological microlattices with different pore characteristics (Supplementary Fig. 16). Overall, the overlapping design strategy can enhance their mechanical robustness. However, not all microlattices follow the rule that the greater the overlap rate, the higher the strength improvement. This is attributed to the limitation of the force acting on the internal strut elements of the microlattice. In addition, it can be found that different topological microlattices can all be used for wood-inspired overlapping design, which verifies the flexibility of our design strategy and the universality of our method, and further enhances the design space of mechanical strength through microlattice topology transformation.

## Transport performance

Transport performance includes air/fluid flow behavior and permeability, the latter of which directly indicates the capacity of microlattices to transport matter and affects matter exchange within structures[52,53]. Permeability is generally related to porosity, pore shape, and pore distribution. In the design and application of biological scaffolds, permeability, as one of the indicators for evaluating biological properties, can reflect the cell proliferation and differentiation of scaffolds[20,21]. The permeability of wood-inspired metamaterial catalysts affects the water purification efficiency of wastewater treatment systems, in which catalytic reaction occurs as wastewater goes through penetrating channels covered by Co/SS[8]. The transport performance of the wood-inspired microlattice metamaterial catalyst is determined mainly by its strut structure and pore characteristics. Although traditional periodic microlattices have multiple geometrical parameters for tailoring permeability, their tunability of transport properties is still limited due to their strong relationships with other performances, such as mechanical performance[20,21,24]. After microlattice overlap for forming the bimodal pores with staggered distribution patterns like the microstructure of Douglas fir, numerous millimeter-scale sub-pores were formed in the wood-inspired metamaterial catalyst. These sub-pores gain more flow channels, functioning as high-throughput transport origins. Besides, the microlattice overlap process generates a higher surface area and smaller pores ($<D_O$, in the traditional periodic microlattices) within the metamaterial catalyst, which provides more contact time for high-efficiency catalysis.

To reveal the fundamental physics of the overlapping strategy on transport performances, we investigated fluid flow behavior inside both the traditional microlattice and wood-inspired metamaterial catalyst using computational fluid dynamics (CFD) finite element calculations, in which a fluid inlet is arranged on the upper surface of the structure perpendicular to the overlapping direction (Fig. 3a and Supplementary Fig. 17). We found a relatively large apparent permeability of traditional microlattices with a 0% overlap rate, which gradually decreased with the increases of the overlap rate. Besides, the lower the inlet velocity, the greater the permeability reduction (Fig. 3b). The permeability of wood-inspired metamaterial catalyst increased with increasing overlap rate and decreased with increasing strut diameter. The wood-inspired overlap design not only regulates the transport properties by adjusting strut diameter but also increases the design of freedom. This means that the strut diameter and overlap rate can synergistically increase the tunability of the transport performance (Fig. 3c). Figure 3d–i compares the distributions of the fluid velocity and shear rate within both traditional microlattices and wood-inspired metamaterial catalysts. Due to the large pore and periodic arrangement of traditional microlattices, the inlet fluid transports quickly without obstacle and induces very large-scale high-velocity (Fig. 3d) in the center of the multilayer pore channels. By contrast, due to the sub-pores and high surface area generated by the microlattice overlapping process, the inlet fluid can only slowly penetrate the wood-inspired metamaterial catalyst, causing much lower velocity in the pores (Fig. 3f). Comparison between Fig. 3g and 3h confirms that friction between the fluidic liquid and the strut surfaces increases the velocity, as well as causes the shear rate concentration in the liquid-solid interface (Fig. 3i). These make the wood-inspired metamaterial catalyst behave lower permeability, which is conducive to the contact between wastewater fluid and metamaterial catalyst, thus improving the reaction efficiency. In addition, we compared the transport permeability of the wood-inspired metamaterial catalysts with that of other structural types such as simple cubic (SC), face-center cubic (FCC), diamond, cuboctahedron, octahedron and octet-truss under the same inlet condition (Fig. 3j). We also calculated the pressure drop (Supplementary Fig. 18), which was the average pressure of the inlet surface at 1 mm/s inlet velocity. The wood-inspired overlapping design strategy facilitates increased pressure drop. Compared to traditional periodic microlattices, our developed wood-inspired metamaterials have higher strength and still good permeability. Besides, overlapping microlattice design enhances the tailoring space for strength and permeability. We also need to note that due to the increase in relative

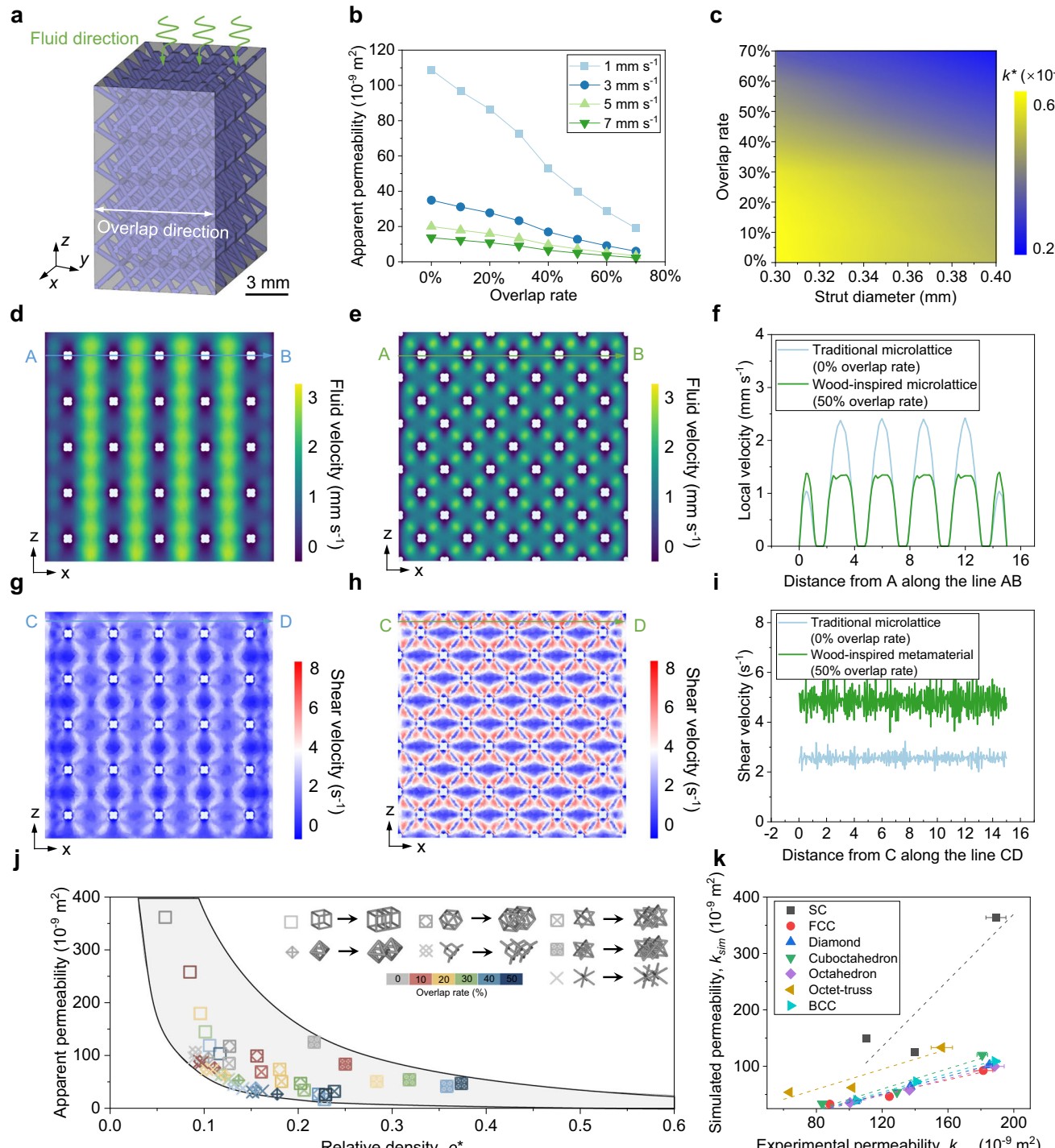

**Fig. 3 | Transport responses of wood-inspired metamaterial catalyst.**
**a** Schematic model of the wood-inspired metamaterial aligned along the overlap direction and the fluid flowed through the top surface. **b** The apparent permeability of wood-inspired metamaterials under different inlet velocities as a function of overlap rates. **c** The contour map of relative permeability as a function of strut diameters and overlap rates. **d**, **e** Simulated distributions of the fluid velocity inside the traditional and wood-inspired metamaterials, respectively. **f** The localized velocity comparison between traditional microlattice metamaterial and wood-inspired metamaterial. **g**, **h** Simulated distributions of the shear velocity inside the traditional and wood-inspired metamaterials, respectively. **i** The localized shear velocity comparison between traditional microlattice metamaterial and wood-inspired metamaterial. **j** Comparison of the apparent permeability of microlattice metamaterials under different overlap rates and structural types as a function of relative density. In the simulation, we assume the magnitude of the inlet velocity is 1 mm/s, the fluid velocity indicates the fluid transport velocity in the pore channels and the shear velocity refers to the change rate of the flow velocity of the fluid relative to the radius of the pore channels. **k** Experimental vs. computational permeability results of the traditional microlattices and different wood-inspired metamaterials. The experimental samples for permeability tests are metamaterial catalysts with 0, 30, and 50% overlap rates and 0.4 mm strut diameter. Error bars represent standard deviation ($n = 3$). Figure 3a–c, f, i–k are provided as a Source Data file.

density, the permeability of the metamaterial structure is inevitably decreased. But for microlattice metamaterials used for structural catalysis, lower permeability means more complete catalytic processes. High permeability and high relative density are the design goals for robust and high-throughput metamaterials for the applications of high-efficient matter transportation without the reaction between liquid and solid. However, we consider that in the application of metamaterial catalysts, having relatively lower permeability and higher relative density within a certain permeability range is the ultimate goal direction.

The permeability of the traditional microlattices and different wood-inspired metamaterials have been experimentally tested, which is presented in Experiments and Methodologies. The experimental permeability results were compared with the corresponding computational permeability results of the traditional microlattice and different wood-inspired metamaterials, which was presented in Fig. 3k. There is usually a certain deviation in the simulated permeability and experimental permeability, but even if the microlattice topology is different, there is still a high linear relationship under different overlapping designs, indicating a good agreement between experiments and simulations. The permeability values of CFD simulations were well correlated ($R^2$ = 0.80, 0.97, 1.00, 0.94, 0.99, 0.90, and 1.00, for SC, FCC, diamond, cuboctahedron, octahedron, octet-truss, and BCC, respectively) with experimental results. The discrepancies in experimental and computational permeability values could be attributed to the following reasons: First, the SLM-printed microlattices have slight surface roughness, leading to the overestimated permeability. Second, the fluid height could be a factor for the constant pressure permeability testing setup, while the height of the pressure head in the laboratory is lower than the optimal value[20]. Finally, the small gap between the permeability testing setup and the periphery of the tested samples also results in permeability discrepancies.

## Water purification capacity and mechanism

In the process of verifying the wood-inspired metamaterial catalyst (mainly Co/SS) activated peroxymonosulfate (PMS) for water purification, the metamaterial catalyst was assembled into a fixed-bed water purification system. The results of the degradation process of sulfamethoxazole (SMX) were obtained via high-performance liquid chromatography (HPLC). The traditional microlattice with 0% overlap rate and wood-inspired metamaterial catalyst with 30, 50, and 70% have all completed the degradation process of SMX within 15 min (Fig. 4a). The kinetics of 20 ppm SMX degradation was presented (Supplementary Fig. 19) and analyzed by a pseudo-first-order reaction model expressed as follows:

$$-\ln\left(\frac{C}{C_O}\right) = kt \tag{1}$$

where $C_O$ is the initial concentration of SMX, $C$ is the concentration of SMX after treatment, and $k$ is the kinetic constant of the reaction. Figure 4b summarizes the dynamic constant results normalized by the occupied space volume, which shows a positive correlation with the overlap rate. These results are all consistent with the transport simulation results described above. The 70% overlap metamaterial catalyst showed the highest reaction kinetic constant per unit volume after normalization (0.191 cm$^{-3}$). Meanwhile, the kinetic constants of wood-inspired metamaterial catalyst activation PMS with different strut diameters were also tested and normalized (Supplementary Fig. 20). With the increase in the strut diameter, the surface area of the metamaterial catalyst was increased, which was accompanied by the increase in the PMS activation rate. To confirm the organic pollutant removal rate, we characterized the organic mineralization rate in the 70% overlap metamaterial catalyst water purification system by measuring the total organic carbon (TOC). The unmineralized organic

pollutants decreased to 24.1% at 15 min and decreased to 18.8% at 30 min, proving that most organic matter is mineralized into non-toxic inorganic matter (Fig. 4c). Meanwhile, high-performance liquid chromatography-mass spectrometry (HPLC-MS) was used to detect intermediates in the degradation process. The mass spectrometry of the possible intermediate products is depicted in Supplementary Fig. 21a. Intermediate products with m/z values of 97, 137, 179, 181, 207, 228, 252, 266, 268, and 282 were detected by the negative ionization mode of the MS spectra. Based on previous reports[54,55], the possible degradation pathways of SMX are depicted in Supplementary Fig. 21b. The toxicity of SMX pollutants and possible degradation intermediates was assessed by a toxicity estimation software tool (T. E. S. T.). The fathead minnow LC50-96 h was investigated by the mode of action method[56]. As shown in Supplementary Fig. 22, SMX with a fathead minnow LC50-96 h value of 40.23 mg L$^{-1}$ was considered "harmful", while the toxicity of intermediates produced in pathway I was increased, while the toxicity of intermediates in pathway II was decreased. Although some toxic intermediates were produced during the degradation process, considering that the Co/SS metamaterial catalyst /PMS system has a strong oxidation and mineralization capacity for SMX, it should be considered as an environmentally friendly water treatment system.

Compared with some other PMS catalytic degradation systems, this degree of mineralization of Co/SS metamaterial catalysts is significant (Supplementary Table 1). The Co/SS metamaterial catalyst/ PMS catalyzed SMX degradation process is beneficial in reducing the toxicity of SMX antibiotics. Most intermediates showed a trend of decreasing toxicity, although a small percentage remained toxic. Therefore, Co/SS metamaterial catalyst as a persulfate catalyst that can highly mineralize SMX should be an effective means of detoxification.

Furthermore, the mechanism of PMS activation by the wood-inspired metamaterial catalyst was demonstrated by using active substance quenching experiments. In this experiment, methanol (MeOH), tert-butyl alcohol (TBA), and furfuryl alcohol (FFA) were equipped as quenching agents. The experimental results of active substance quenching are shown in Supplementary Fig. 23a. The addition of each of the three quenching agents significantly reduced the catalytic rate, confirming that $\cdot$OH, $SO_4^{\cdot-}$, and non-radical active substances occurred in the degradation system. To quantify the contribution of different radicals, we fitted the reaction kinetic constants in the presence of different quenching agents using Eq. (1) (Supplementary Fig. 23b). When added in sufficient abundance, the quenching agent will immediately deplete the corresponding active substance; thus, the contributions of different active substances to the degradation process can be quantified. The results showed that $\cdot$OH, $SO_4^{\cdot-}$, and non-radical active substances accounted for 46.3, 25.5, and 28.6% of the degradation process, respectively (Fig. 4d). Therefore, the degradation process was a free radical pathway dominated by $\cdot$OH.

Electron paramagnetic resonance (EPR) spectroscopy is widely used to verify the existence of free radicals. The technique can directly detect the active components in the degradation system. The reagents 5,5-dimethyl-1-pyrroline-N-oxide (DMPO) and 2,2,6,6-tetramethylpiperidine (TEMP) were added to the degradation system to detect potential free radicals. The EPR results are shown in Fig. 4e. In the system with DMPO as the collector, the EPR signal of the DMPO-OH adduct ($\alpha$N = $\alpha$H = 14.8 G) was detected. The EPR signal of the DMPO-SO$_4$ adduct ($\alpha$N = 13.2 G, $\alpha$H = 9.6 G, $\alpha$H = 1.48 G, $\alpha$H = 0.78 G) was not detected because of the low concentration of $SO_4^{\cdot-}$. In the system with TEMP as the trapping agent, TEMPO ($\alpha$ = 16.9 G) exhibited a characteristic threefold signal of 1:1:1, indicating generation. Therefore, the EPR results demonstrate that the non-free radical pathway includes $^1O_2$ as the active substance. To verify electron transfer during degradation, a time-current curve was performed under the open circuit voltage (Fig. 4f). When only a metamaterial catalyst exists in the system, there is no obvious current response. With the addition of PMS, the current

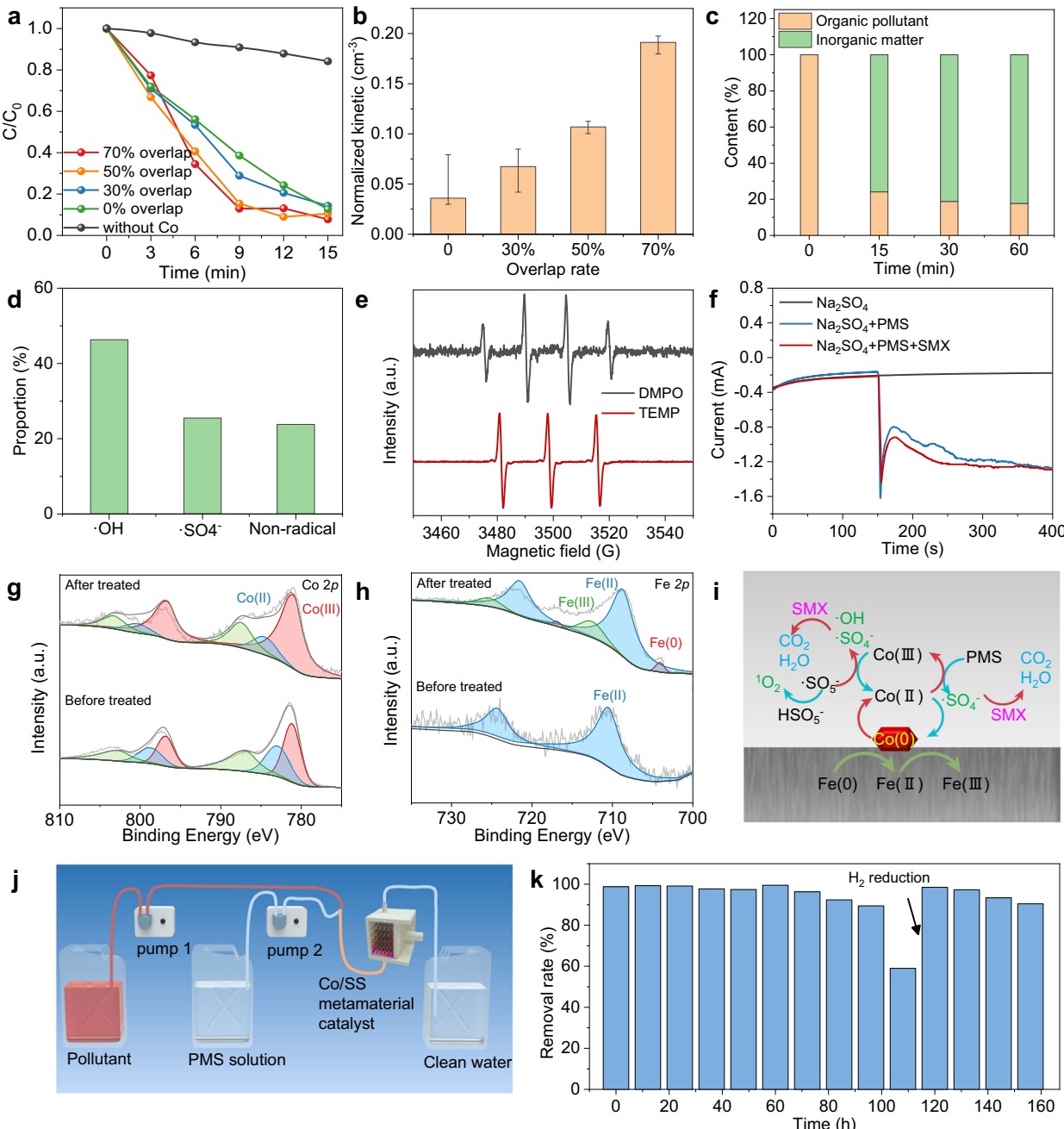

**Fig. 4 | Water purification capacity and catalytic mechanism of wood-inspired metamaterial catalyst. a** The degradation results of SMX by different metamaterial catalysts. **b** Normalized degradation kinetics constant of wood-inspired metamaterial catalysts with different overlap rates. The error bars present the upper and lower limits of the normalized kinetic constant for multiple experimental testing. **c** The organic mineralization rate in the 70% overlap metamaterial catalyst water purification system by measuring the total organic carbon (TOC). **d** The ratio of different active substances obtained by free radical quenching experiment. **e** EPR signal of the active substances captured by DMPO and TEMP. **f** current-time curves of the different reaction systems. XPS characterization of **g** Co and **h** Fe in wood-inspired metamaterial catalysts before and after water treatment. **i** Proposed mechanism of PMS activation by wood-inspired metamaterial catalysts and the strong synergy between Co and Fe. **j** Schematic diagram of long-time water treatment device. **k** Long-term performance of Co/SS metamaterial catalyst for SMX degradation. Figure 4a–h, k are provided as a Source Data file.

density increased significantly, indicating that the interaction between PMS and Co/SS metamaterial catalyst formed a metastable complex. In the system with both PMS and SMX, the current density was further enhanced, implying electron transfer from SMX to Co/SS metamaterial catalyst/PMS complex, which proves that direct electron transfer also exists in the degradation process of the non-radical pathway.

To find out the change in the valence state of metamaterial catalyst surface elements during degradation, X-ray photoelectron

spectroscopy (XPS) was investigated. Before the reaction, the Co/SS metamaterial catalyst exhibited Co $2p_{3/2}$ peaks at 781.3, and 783.1 eV, corresponding to Co(III) and Co(II), respectively, and the peak area ratio $I_{CoII}:I_{CoIII} = 0.32$. After the reaction, the peak area of Co(III) increased significantly, and $I_{CoII}:I_{CoIII} = 0.21$ (Fig. 4g). These results indicate the conversion of Co(II) and Co(III) on the Co/SS metamaterial catalyst surface during PMS activation and the catalytic reaction. Moreover, the Co/SS metamaterial catalyst before the reaction

exhibited Fe $2p_{3/2}$ peaks at 710.6 eV, indicating that the frame material had surface oxidation before the water treatment (Fig. 4h). After the water treatment, the Co/SS metamaterial catalyst featured a Fe $2p_{3/2}$ peak at 712.5 eV, indicating the generation of Fe(III). According to the changes in the valence state, we propose the following mechanism of PMS activation by Co/SS metamaterial catalyst (Fig. 4i):

$$\equiv Co(0) + 2HSO_5^- \rightarrow \equiv Co(II) + 2HSO_4^- + 2\cdot OH \tag{2}$$

$$\equiv Co(II) + HSO_5^- \rightarrow \equiv Co(III) + \cdot SO_4^- + OH^- \tag{3}$$

$$\equiv Co(III) + HSO_5^- \rightarrow \equiv Co(II) + \cdot SO_5^- + H^+ \tag{4}$$

$$\equiv Co(III) + \cdot SO_5^- + H^+ \rightarrow \equiv Co(II) + \cdot SO_4^- + \cdot OH \tag{5}$$

$$\equiv Fe(0) + \equiv Co(II) \rightarrow Co(0) + \equiv Fe(II) \tag{6}$$

$$\equiv Fe(II) + \equiv Co(III) \rightarrow 2Co(II) + \equiv Fe(III) \tag{7}$$

$$HSO_5^- + \cdot SO_5^- \rightarrow {}^1O_2 + H^+ + 2SO_4^{2-} \tag{8}$$

The corrosion resistance of Co/SS metamaterial catalysts is the basement for long-term use. The corrosion resistance of the Co/SS framework has been reduced to a certain extent; this is the response for Co as a reactive substance is not as resistant to corrosion of PMS as Fe (Supplementary Fig. 24a–c and Supplementary Note 6). According to the standard electrode potential $E^\ominus$ of Fe/Fe$^{2+}$ is −0.44 V and $E^\ominus$ of Co/Co$^{2+}$ is −0.28 V, which means that galvanic corrosion occurs when Co/SS contact, and electrons are transferred from Fe to Co. This result hurts the corrosion resistance of SS-based metamaterials as shown in the above corrosion data. However, electron injection into Co is conducive to the circulation of Co species in the catalytic process (Supplementary Fig. 24d).

The long-term stability of the wood-inspired metamaterial catalyst is also an important factor in practical utilization. To understand the maximum stability of Co/SS metamaterial catalysts, a water treatment system as shown in Fig. 4j, was assembled, which consisted of two peristaltic pumps mixing 20 ppm SMX and 0.4 g L$^{-1}$ PMS at a flow rate of 0.5 mL min$^{-1}$ respectively into a fixed-bed processor equipped with Co/SS metamaterial catalysts, and a third bucket was used to collect the treated liquid. It was found that the Co/SS metamaterial catalysts could run continuously for more than 96 h and the degradation efficiency remained above 90%, as shown in Fig. 4k. After 108 h treatment, the degradation efficiency decreased, which may be due to the occurrence of surface deactivation and passivation. This deactivation is attributed to the lack of active Co species, and the degradation efficiency can be restored to 98% by Ar/H$_2$ reduction at 300 °C for 3 h. It should be noted that a vast of AOP-activated materials have only studied the service life of five cycles, and this stability is far from enough for practical water treatment materials. Our Co/SS metamaterial catalysts show the stability of the day level, which has a certain positive significance for the practical application of water treatment materials. Compared with other high-stability water treatment systems, the stability of our system has a considerable advantage (Supplementary Table 2).

## Practicality, designability, and applicability

Actual water bodies contain numerous kinds of inorganic anions and other organic substances, many of which will inhibit PMS activation. We verified the resistance of the wood-inspired metamaterial catalyst to 200 ppm NaHCO$_3$, Na$_2$SO$_4$, NaNO$_3$, NaCl, and 20 ppm humic acid (HA) in the PMS activation process. Although impurities influenced the

$\cdot$OH-dominated activation process (Fig. 5a), the Co/SS metamaterial catalyst still achieved over 93% treatment efficiency within 15 min in 200 ppm Na$_2$SO$_4$, NaNO$_3$, and NaCl. To verify the effect of pH, the wood-inspired metamaterial catalyst was placed in environments of various pH, from 3 to 9, and they maintained excellent SMX treatment efficiency under both acidic and neutral conditions (pH = 3, 5, and 7, Fig. 5b). The treatment efficiency was inhibited under alkaline conditions (pH = 9), but a 75% degradation rate was still achieved after 10 min. It could be that OH$^-$ covers the surface of the catalyst, resulting in unfavorable adsorption of PMS and pollutants. After that, we used actual water samples as the solvent to test the activation degradation of SMX by PMS. The results showed that the degradation rate of the wood-inspired metamaterial catalyst reached more than 95% after 15 min in tap water and Yangtze River water as the solvent (Supplementary Fig. 25), which indicated the application potential of Co/SS metamaterial catalyst in the real environment. The wood-inspired metamaterial catalyst-activated PMS system can not only be used for SMX degradation but also bisphenol A (BPA) and norfloxacin (NOR) can achieve 95.3 and 98.2% efficiency within 15 min, respectively (Fig. 5c). This indicates that the wood-inspired metamaterial catalyst-activated PMS water purification system can be used in water samples with complex contaminants.

The above results demonstrated the practicality of the wood-inspired metamaterial catalyst. On this basis, we elucidated the geometrical characteristics of wood-inspired metamaterial catalysts (Fig. 5d). The structural design (mechanical and transport properties) and catalytic performances (treatment duration) of the wood-inspired metamaterial catalyst can be leveraged to obtain a large-scale, continuous-flow, and high-throughput water treatment, demonstrating its application potential in flow-related engineering (Fig. 5e). We can also achieve desirable throughput and water quality by flexibly and precisely tuning the structure and catalytic performances. The stiffness ($E$) of the wood-inspired metamaterial catalyst relies on the strut diameter ($d$) and overlap rate ($\delta$). The water flow time ($t_1$) depends on the surface porosity ($S_A$) and the overall length ($N \times a$), while the catalytic time ($t_2$) is related to the permeability and the type of catalyzer ($x$). In addition, the permeability and surface porosity are both influenced by geometrical characteristics, including strut diameter and overlap rate. This relationship network involves two constraints. First, the water flow time needs to be shorter than the catalytic time to ensure complete purification of the sewage. Second, the impact strength of water flow needs to be less than the rigidity of the wood-inspired metamaterial catalyst. A reverse engineering-guided design was conducted according to the established relationship and known constraints related to geometric topology, and physical and chemical properties. The stiffness, transport property, and catalytic capacity of the wood-inspired metamaterial catalyst were tunable, which indicates that the structure-function integrated manufacturing of water purification systems can be easily realized through metamaterial structural design and 3D printing. Microlattices or porous structures have strong structural designability and additively manufacturability[57], while also exhibiting significant mechanical robustness for metal-based materials in 3D printing component manufacturing. However, due to the low selectivity of metal-based materials in the SLM process, many materials with good catalytic performance are difficult to manufacture as components, which greatly limits their catalytic applications in the fields of environment and energy. On the contrary, metal-organic frameworks or carbon-based synthetic materials have ultra-high surface area and permanent porosity, making them ideal candidates for fuel (hydrogen and methane) storage, carbon dioxide capture, and catalytic applications[58]. However, they are still full of challenges for large-scale engineering applications in terms of the current manufacturable sizes of only nanometers/micrometers and poor chemical stability. Compared with the microlattice/porous structure and synthesis catalyst, the wood-inspired metamaterial catalyst, which was fabricated by the

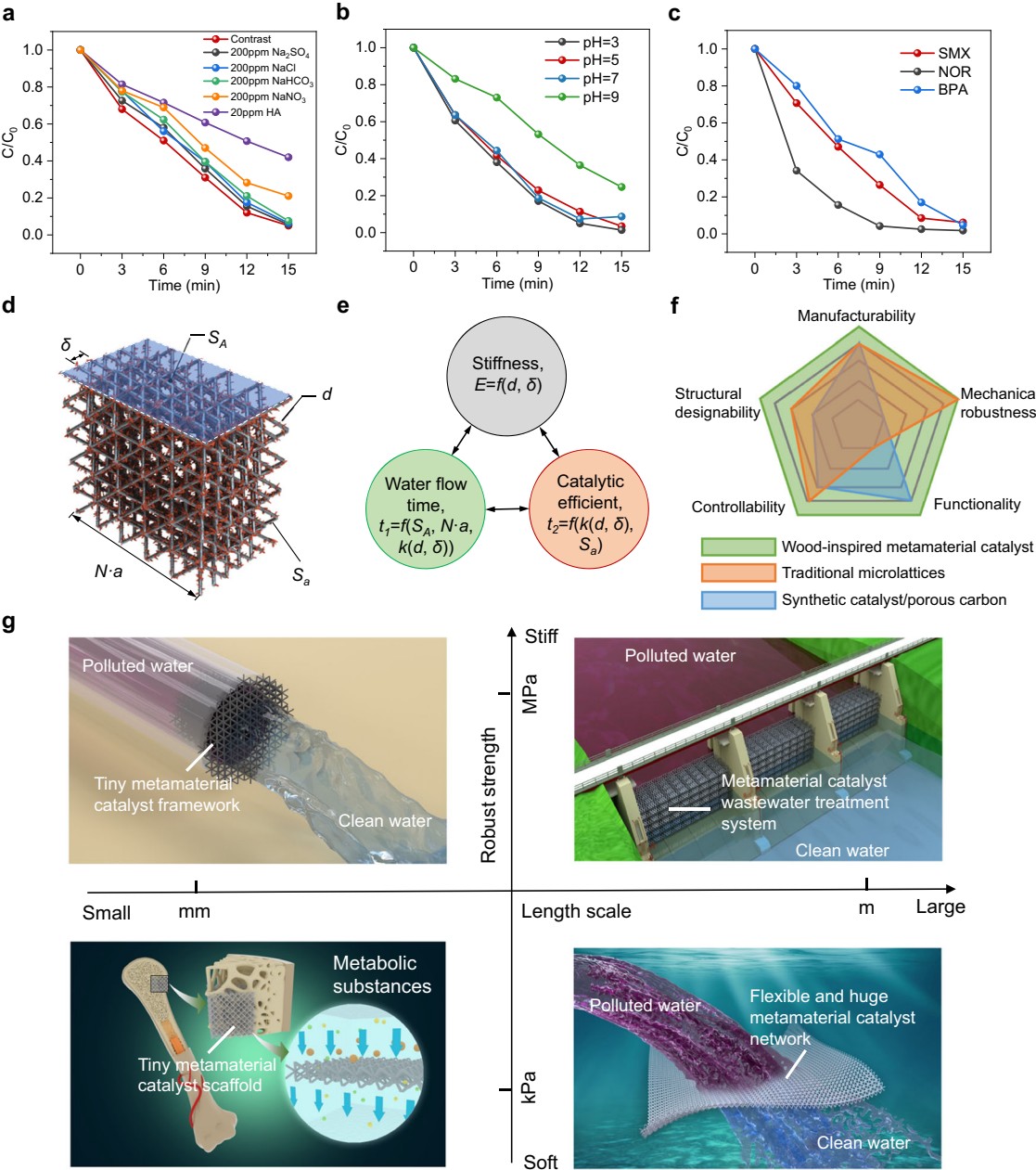

**Fig. 5 | Robustness, designability, and applicability of the wood-inspired metamaterial catalyst.** The water treatment capacity of the wood-inspired metamaterial catalyst **a** under different anion and HA interferences, **b** at pH from 3 to 9 and interval 2, and **c** in different contanmicates. **d** Geometrical characteristic of wood-inspired metamaterial catalyst. **e** Design paradigm of wood-inspired metamaterial catalyst systems for water purification according to topological geometry, physical properties, and chemical properties. **f** General characteristics comparison of the common materials used for water purification (more details presented in Supplementary Table 3). **g** Possible applications of the wood-inspired metamaterial catalyst at different length scales, ranging from tiny medical scaffolds and small waterpipes (left) to scalable, flexible architectures (right). The application of water purification capacity is based on this work while the metamaterial-based scaffold inside the bone can conduct chemical reactions to harmless excretory waste (left lower corner). Figure 5a–c are provided as a Source Data file.

simple preparation through the combined process of metal 3D printing and electrochemical deposition, integrates the dual functions of structural strength and functional catalysis, has strong manufacturability, higher macro- and micro- structural designability, stronger controllability of geometry and performance, higher strength and high-throughput water purification capability (Fig. 5f).

The wood-inspired metamaterial catalyst composed of assembled microlattices can be realized at different length scales (Fig. 5g), because of the advances in 3D printing technologies, which could, in principle, scale the pore size from the micrometer to the meter scales,

and use different constitutive materials, targeting different applications (such as different water purification scenes). Meanwhile, it is even possible to consume harmful metabolic substances by implanting tiny metamaterial scaffolds for high-throughput catalytic reactions in organisms (Fig. 5g). Our work focuses on high-throughput structural design to achieve robust and efficient metamaterial catalysts. There are certain limitations in the manufacturing process selection and material selection. However, the design strategy of imitating wood bimodal pores and the strategy of coupling 3D printing with electrochemical deposition to form high-throughput metamaterial catalysts

can be used for reference in other flow-catalytic fields. Besides, compared with the original performance of Douglas fir wood, the 3D-printed wood-like overlapping microlattice metamaterial still exhibits moderate mechanical and excellent transport properties in the macroscale (Supplementary Note 7 and Supplementary Figs. 26–28).

In summary, we report a structure-function integrated wood-inspired metamaterial catalyst water purification system based on metal 3D printing technology and an electrochemical deposition process. The wood-inspired metamaterial structure was developed by a microlattice overlap strategy, which allows the creation of numerous sub-pores between the microlattice units with increased robustness and surface area per volume. We presented that a wood-inspired metamaterial catalyst with a 70% overlap rate has three times more strength, three times surface area per unit volume, and four times normalized reaction kinetics than traditional microlattices. The overlap strategy and subsequently formed bimodal pores greatly improve the robustness of wood-inspired metamaterial catalysts. The overlap-generated numerous sub-pores and high surface area make the inlet fluid slowly penetrate the structure, causing much higher velocity and adequate contact reaction in the pores within wood-inspired metamaterial catalysts. Compared with multi-process and time-consuming water purification systems, the metamaterial catalysts possess higher efficiency, low cost, and scalability. These wood-inspired metamaterial catalysts with superior mechanical robustness, high-throughput flow, and high-efficiency catalysis can replace traditional water purification systems, as well as inspire unprecedented development in the fields of flow catalysis and other structure-function integrated applications.

## Methods

### Material preparation and metamaterial fabrication

Several chemicals, including cobalt chloride hexahydrate ($CoCl_2 \cdot 6H_2O$, 99.9%), sulfamethoxazole (SMX, 98%), norfloxacin (NOR, 98%), bisphenol A (BPA, 99.0%), peroxymonosulfate ($2KHSO_5 \cdot KHSO_4 \cdot K_2SO_4$, 98%), methanol (MeOH, 99.9%), furfuryl alcohol (FFA, 98%), tertiary butyl alcohol (TBA, 99.5%), ethanol (EtOH, 99.8%), sodium sulfate anhydrous ($Na_2SO_4$, 99%), sodium nitrate ($NaNO_3$, 99%), sodium bicarbonate ($NaHCO_3$, 99.5%), sodium chloride (NaCl, 99.5%), humic acid (HA, 90%), sodium thiosulfate ($Na_2S_2O_3$, 99%), boric acid ($H_3BO_3$, 99.5%), hydrochloric acid (HCl, 36%), 5,5-dimethyl-1-pyrroline-N-oxide (DMPO, 97%), and 2,2,6,6-tetramethylpiperidine (TEMP, 97%), were purchased from Macklin (Shanghai, China). All solvents and reagents were obtained from commercial sources and were used without further purification.

The raw materials of a commercial gas-atomized 316 L powder were used for SLM. The 316 L powder had the following particle geometry features: $D_{10}$: 22.3 μm, $D_{50}$: 32.8 μm, and $D_{90}$: 47.0 μm, which met the SLM process requirements. The unit cell size of microlattice metamaterials with different diameters and overlap rates was kept constant at 3 mm. Each microlattice comprised five unit cells along each Cartesian direction. All microlattice samples were additively manufactured using an HK125 SLM machine at an input laser power of 320 W, scanning speed of 650 mm/s, layer thickness of 50 μm, and hatch distance of 140 μm, and the scanning direction was alternately rotated by 67° between adjacent layers. For each microlattice topology, three identical samples were additively manufactured.

Synthesis of Co/SS metamaterial catalyst: As prepared 316 L metamaterial structure is soaked in 1 M HCl solution for 15 min to remove the surface oxide layer, followed by three rinses in DI water. Subsequently, Co coating was in situ electrochemically deposited on the 316 L metamaterial structure. The electrolyte was prepared by adding 0.3 M $Co(NO_3)_2 \cdot 6H_2O$, and 0.5 M $H_3BO_3$ in the electrodeposition bath. A three-electrode system was constructed by using the saturated Ag/AgCl reference electrode, the graphite rod as a counter electrode, and the working electrode. Next, a voltage of −1.4 V vs. Ag/AgCl was supplied continuously on nickel foam for 300 s to prepare

the Co/SS metamaterial catalyst. Then, the electrolyte attached to the catalyst surface was rinsed with DI water and EtOH.

### Morphology characterization and compression tests

Morphological observation images were obtained via SEM (JSM-7600F, Japan). The SLM-induced additive geometric deviations of microlattice metamaterials were quantitatively analyzed via micro-focus computed tomography (micro-CT) using a GE Phoenix X-ray system. These data were further analyzed and visualized using the commercially available image analysis software VG Studio Max 3.4 (Volume Graphics GmbH, Heidelberg, Germany). A series of uniaxial quasi-static compression tests were performed at a loading rate of 1.2 mm/min using an AG-IC100 KN electronic universal testing machine (Shimadzu, Japan) at room temperature. To elucidate the mechanical responses of the microlattices, Young's modulus, yield strength, and compressive strength were determined from stress–strain curves following ISO 13314:2011.

### Finite element model and permeability tests

A laminar CFD model was constructed to study the fluid permeability and mass transport behavior in *Comsol Multiphysics* 5.3a software. The fully developed fluid permeability behavior was studied using the Navier–Stokes equation for an incompressible fluid with a constant density and viscosity.

$$\rho \frac{\partial v}{\partial t} - (v \cdot \nabla)v + \frac{1}{\rho}\nabla P - \mu\nabla^2 v - F = 0, \nabla \cdot v = 0 \tag{9}$$

where $\rho$ is the density of the transport fluid ($kg/m^3$), $v$ is the fluid velocity (mm/s), and $\mu$ is the dynamic viscosity coefficient of the fluid (Pa s); $\nabla$ and $P$ are the del operator and pressure (Pa), respectively; and $F$ denotes other forces (gravity or centrifugal force; in this case, $F = 0$).

Water was used as the flow medium. The velocity of the inlet-flow side was 0.001 m/s, and the outlet pressure of the outlet-flow side was set as zero. The gradient drop, also called pressure drop, was obtained using the equation:

$$\Delta P = P_{in} - P_{out} \tag{10}$$

To evaluate the transporting capacity of microlattice scaffolds, computational permeability was calculated using the following formula:

$$k = \frac{v \cdot \mu \cdot H}{\Delta P} \tag{11}$$

where $H$ is the height of the CFD model. The physical properties of water with a density of 1000 $kg/m^3$ and a viscosity of $1.01 \times 10^{-3}$ Pa s were considered in the medium material model.

The fluid permeability is determined by applying Darcy's law, making sure the Reynolds number ($Re$) remains sufficiently low (typically $Re < 10$). The adopted permeability setups make use of a falling-head approach, where a tall column of water is placed above the porous structure and is allowed to flow through it under the action of gravity[59]. The same method has also been presented in other literature on permeability testing of porous structures[20,23,60,61]. By using Darcy's law, $v = K\lambda$, the experimental permeability of 3D-printed microlattices can be obtained, in which $v$, $K$, and $\lambda$ are the fluid velocity, hydraulic conductivity, and hydraulic gradient, respectively. The initial liquid height $H_1$ and the final liquid height $H_2$ were captured at $T_O$ and $T_i$, respectively. The hydraulic gradient, $\lambda$, is described by:

$$\lambda = \frac{H_1 - H_2}{H} \tag{12}$$

The hydraulic conductivity, $K$, can be obtained by integrating the water flow rate over the measurement time $T_i - T_O$, as following:

$$K = \frac{aH}{A(T_i - T_0)} \ln\left(\frac{H_1}{H_2}\right) \tag{13}$$

where $a$ and $A$ are the cross-section areas of the standpipe and the samples, respectively. The experimental permeability of the 3D-printed microlattice samples can be calculated by:

$$k_{\text{exp}} = \frac{K\mu}{\rho g} \tag{14}$$

## Degradation process

For the pollutant degradation experiment, the reaction system was equipped with a 100 mL beaker containing SMX ($20\,mg\,L^{-1}$) and PMS ($0.2\,g\,L^{-1}$). The reaction solution was pumped into the Co/SS-based metamaterial fix-bed system, and the flow rate was $12\,mL\,min^{-1}$. For catalytic rate calculation, the reaction liquid was returned to the original beaker. During the stability test, two peristaltic pumps mixed 20 ppm SMX and $0.4\,g\,L^{-1}$ PMS at a flow rate of $0.5\,mL\,min^{-1}$, respectively, and the liquid did not flow back to the original beaker after the reaction. The collected post-reaction liquid (1 mL) was added to a centrifuge tube containing ethanol (0.5 mL) to quench the reaction. The pollutant was analyzed using high-performance liquid chromatography (HPLC, Agilent 1100, USA). $Na_2S_2O_3$ was used as quencher in TOC (TOC-L, shimadazu, Japan).

In the free radical quenching experiment, 0.5 M MeOH, 0.5 M TBA, and 0.03 M FFA were used as trapping agents, respectively. The existence of $^-OH$, $SO_4^-$, and $^1O_2$ were verified via the electron paramagnetic resonance (EPR) spectrum. DMPO and TEMP were used as capture agents to detect $^-OH + SO_4^-$ and $^1O_2$, respectively. The molecular debris during degradation was detected by HPLC-MS (290-6530-Qtof, Agilent, USA).

## Electrochemical measurement

The electrochemical workstation (CORRTEST CS6) was used to perform a chronocurrent curve (i-t) to measure the electron transfer direction between PMS/ catalyst/pollutant using a three-electrode system consisting of a graphite counter electrode and a saturated Ag/AgCl reference electrode, 0.5 M $Na_2SO_4$ solution as electrolyte. The PMS solution or the mixed solution of PMS and SMX was injected after scanning for 30 s at the open potential, the mixing concentration of PMS was $0.2\,g\,L^{-1}$, and the concentration of SMX solution was 20 ppm. The polarization current data were recorded at a scan rate of $5\,mV\,s^{-1}$ in $0.2\,g\,L^{-1}$ PMS solution or DI water from −0.8 to +1.6 V vs. Ag/AgCl after 30 min open current potential (OCP) scanning.

## Data availability

The data generated in this study are provided in the Source Data file. The dataset for this work is available at Figshare, ref. 62. Source data are provided with this paper.

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

## Acknowledgements

B.S. acknowledges the National Natural Science Foundation of China (No. 52101255), the Key-Area Research and Development Program of Guangdong Province (No. 2020B090923001), the Academic Frontier Youth Team at the Huazhong University of Science and Technology (No. 2018QYTD04), and the National Key R&D Program of China (2021YFA1202300). Y.Y. acknowledges the National Natural Science Foundation of China (No. 52275331, 52371223). J.L. acknowledges the Shenzhen-Hong Kong Science and Technology Innovation Cooperation Zone Shenzhen Park Project: HZQB-KCZYB-2020030. L.Z. acknowledges the National Natural Science Foundation of China (No. 52305360), and the Hong Kong Scholars Program (No. XJ2022014). The authors thank the Analytical and Testing Center of HUST for the SEM examination and the State Key Laboratory of Materials Processing and

Die & Mould Technology for compression tests. The authors thank the Shiyanjia Lab (www.shiyanjia.com) for the XPS, HPLC-MS, and TOC analysis.

## Author contributions

L.Z. and H.W.L. contributed equally. L.Z., H.W.L., B.S., Y.Y., and J.L., conceived the ideas and designed the research; L.Z. performed the mechanical experiments and FE simulation. L.Z., H.L., X.W., J.F., and Z.Z. printed the metamaterial samples. H.W.L. and S.Z. investigates the microstructure of wood and the catalytic properties of samples. J.G., L.L., G.L., and W.S. participated in the analysis of experimental data. L.Z., H.W.L., B.S., and Y.Y. drafted the manuscript; B.S., P.W., Y.Y., Y.S., and J.L. revised the manuscript; B.S., Y.Y., and J.L. supervised the project. All authors participated in the discussion of the results.

## Competing interests

The authors declare no competing interests.
