## [Peer Review File · Nature Communications]

Wood-inspired metamaterial catalyst for robust and high-throughput water purificationREVIEWER COMMENTS

Reviewer #1 (Remarks to the Author):

The authors presented 3D-printed mechanical metamaterials for water purification application. In particular, to achieve robust and high-throughput water purification, the authors designed Douglas fir wood-inspired metamaterial catalysts. With appropriate experimental and simulation data, the concept was successfully demonstrated. Overall the findings look interesting and I would like to suggest several major comments to consider the acceptance of the paper.

1. In the mechanical robustness analysis, analysis of weight was omitted. How light a device can be made should also be considered.
2. Technology review is needed from various perspectives such as weight, strength, price, structure differentiation, and mechanism. Please refer to below papers as prior arts.
"Ultralight Metallic Microlattices", Science 334, 962, 2011
"Strong, lightweight, and recoverable three-dimensional ceramic nanolattices", Science 345, 1322, 2014
"Resilient 3D hierarchical architected metamaterials", PNAS 112, 11502, 2015
3. Please confirm the discrepancy between simulation and experimental results of strain-stress curve (figure 2j).
4. Please consider and elaborate elastic deformation and failure modes like buckling and fracture. Also, please provide the maximum stress that the structure can withstand.
5. Please compare the mechanical robustness and transport performance with wood that you referred to. How much can those characteristics be improved when you use metamaterials?
6. Figure 3j, please highlight the ultimate goal direction, current state-of-the-art 3D printed structures and the developed metamaterial device.
7. Provide corresponding experimental results for permeability with different shapes of microlattices. (e.g. fluid velocity)

Reviewer #2 (Remarks to the Author):

Comments to the Authors:

Lei Zhang et al. present a wood-inspired metamaterial catalyst design and structure for water purification. While the main structure is fabricated with 316L stainless steel using metallic 3D printing, the surface area of the structure is doped with cobalt using electrochemical deposition. This approach significantly improves throughput compared to producing the same structure solely using metallic 3D printing. The resulting structures exhibit robustness and efficient permeability for water purification. The paper is well-written and well-presented. The overall utilization of metallic 3D printing and cobalt electrodeposition is a novel approach for water purification. However, there are certain aspects that require further discussion and context in relation to similar technologies.

The paper does not sufficiently elaborate on the advantages and disadvantages of the presented work compared to similar technologies. It is crucial to highlight the advantages to enable researchers to select appropriate methods for different applications. Similarly, it is important to identify the disadvantages to guide future researchers in focusing on areas for technological improvement. While some of these advantages and disadvantages may have been discussed in the authors' prior paper, there is ample opportunity to delve into the trade-offs presented by the metamaterial designs and structures. Furthermore, it would have been beneficial to fabricate and include more complex structures for a comprehensive analysis. I recommend that the authors consider attempting to fabricate such structures or at the very least provide an explanation within the text as to the feasibility of producing more intricate designs (potentially within the conclusions section).

Below are specific comments, suggestions, and questions that need to be addressed prior to the paper's publication in Nature Communications:

1. a. Considering the innovative approach of creating a wood-based metamaterial structure using stainless 316L through 3D printing, it might be beneficial for the authors to delve further into a comprehensive comparison with analogous technologies that possess the potential for heightened efficiency. Have the authors explored contrasting techniques, including those employing alternative materials such as glass, polymers, and examined their respective advantages and drawbacks? [Patrik Schurch et al., Direct 3D microprinting of highly conductive gold structures via localized electrodeposition, doi.org/10.1016/j.matdes.2023.111780 ; X. Tan et al., Single-Step-Lithography Micro-Stepper Based on Frictional Contact and Chiral Metamaterial, doi.org/10.1002/sml.202202128 ; Grabulosa A. et al, Combining one and two photon polymerization for accelerated high performance (3 + 1)D photonic integration, doi.org/10.1515/nanoph-2021-0733]. Moreover, could a more in-depth exploration of comparable methods shed light on the feasibility and potential advantages of utilizing different materials in achieving similar structural metamaterial outcomes?
- b. Given the authors' inclusion of a material comparison table in the supplementary document, which discusses various materials for metamaterial fabrication, could the authors consider integrating relevant insights from this comparison into the main text? This would provide readers with a clearer understanding of the merits of utilizing stainless 316L in their wood-based structure compared to other materials. Additionally, should the authors choose to delve into alternative materials like polymers, could they elucidate the specific attributes that these materials offer and allow readers to identify whether polymers, or metal-organic have the potential to offer performance enhancements for the metamaterial structure [Xueyan Chen et al., Optimal isotropic, reusable truss lattice material with near-zero Poisson's ratio, doi.org/10.1016/j.eml.2020.101048; Furukawa H et al. The chemistry and applications of metal-organic frameworks. *Science* 2013;341:974–86], particularly when compared to stainless 316L?

2. The authors have conducted a comparison between two distinct structures: The first structure features a traditional microlattice consisting of uniform squares with thin walls composed of inclined struts and robust nodes. In the second structure, a unique approach is taken by superposing microlattices that alter the spatial shape of the internal pore region. Remarkably, the technique of overlapping these microlattices emerges as a promising strategy. This approach, encompassing both large and small pores, exhibits the potential to enhance the overall stiffness of the structure. By reinforcing its ability to withstand the impact of fluid flow, this innovative approach holds significant promise for various applications.

Considering the success of this technique, it could be intriguing to explore simulations, fabrication, or

comparative analyses involving structures that employ diverse pore shapes beyond the conventional squares [Huang Y, Lu X, Liang G, Xu Z. Pentamodal property and acoustic band gaps of pentamode metamaterials with different cross-section shapes. *Phys Lett A* 2016;380:1334–8 ; Moein Shafia, Pentamodes : the role of unit cell base topology on mechanical properties, <https://doi.org/10.48550/arXiv.2305.15419>]. Delving into alternative pore geometries, and their corresponding effects on mechanical properties, could provide invaluable insights. For instance, investigating how pore shapes other than squares influence stiffness, resilience, and resistance to fluid flow impact could contribute to an enhanced understanding of how metamaterial structures can be tailored to specific engineering requirements. Such exploration could pave the way for an extended range of applications and further refinement of metamaterial design principles.

3. In the scope of utilizing a stainless steel structure as a substrate for cobalt electrodeposition in water purification applications, have the authors thoroughly investigated the susceptibility of the stainless steel to corrosion phenomena in the intended operational environment? Considering the potential exposure to aqueous solutions and diverse ionic species, have the paper's authors explored how the inherent corrosion resistance of stainless steel could interact with the electrodeposited cobalt layer, both in terms of synergistic corrosion protection and the possibility of galvanic corrosion [Kong, D. et al., Corrosion of metallic materials fabricated by selective laser melting, doi.org/10.1038/s41529-019-0086-1]? Furthermore, have the authors elucidated how the cobalt deposition process can be harnessed not only for catalytic water purification but also as a means of enhancing the overall corrosion resistance of the stainless steel structure when exposed to the operational conditions of a water treatment setting?

4. Given the intricate nature of the structure, it's imperative to delve into potential concerns surrounding cracking within this innovative framework. Moreover, the influence of external forces in real-world scenarios prompts an examination of how these forces could impact the microstructure's stability. In light of these considerations, gaining insights into the following dimensions would undoubtedly enrich the paper's depth:

a. Regarding Potential Cracking: The utilization of stainless steel within the 3D-printed metamaterial structure raises concerns about the propensity for cracks. Has the microstructure been meticulously characterized to identify the presence of existing cracks or anticipate cracks that might manifest during the manufacturing process? Specifically, how has the paper examined the potential influence of incomplete powder fusion during the solidification process on the formation of cracks within the stainless steel microstructure? Moreover, could you elucidate how these considerations are interconnected with the development of the distinctive wood-based metamaterial structure [T. DebRoy et al., Additive manufacturing of metallic components – Process, structure and properties, doi.org/10.1016/j.pmatsci.2017.10.001] ?

b. Addressing External Load Effects: The operational reality of the metamaterial involves encountering external forces, notably water flux in the context of water purification. How has the paper accounted for the impact of such external loads on the structural integrity and stability of the stainless steel microstructure? Additionally, considering the significance of the chosen overlap rates, I'm intrigued to understand whether varying external load conditions could give rise to different modes of crack separation. Can you shed light on the interplay between the microstructure's mechanical response, the influence of water flux, and the potential diversification of crack propagation mechanisms, contingent on the selected overlap rates [N. Shamsaei, A. Yadollahi, L. Bian, S.M. Thompson, *Addit. Manuf.* 8 (2015):

p. 12-35, doi.org/10.1016/j.addma.2015.07.002]?

5. Minor Comments

Page 3, line 57, "transportataion" -> "transportation"

Page 4, line 97, "achievinge" -> "achieving"

Reviewer #3 (Remarks to the Author):

This manuscript reports a new way of creating water purification devices using metamaterials design. The structural configurations with highly stable material properties generated by the wood-inspired design can potentially be important.

Positively, this is highly relevant and highly unique to the current water treatment practices. It will really have an impact on how we manufacture water treatment devices. The bottom up approach of designing the device using 3D printing and including catalysts in them will significantly enable the new industrial manufacturing of water filtration devices.

However, the major concern is that why the authors are calling the 3D printed structure a metamaterial while all the other papers mentioned in the manuscript (showing comparative) results didn't although all of these recent papers and this manuscript are using similar 3D printing approach.

The authors suggested that the metamaterial structure help to slow down the water permeability. I am not sure if that is a desirable output for high volume water industry. If the authors can justify the application space it would be great.

The authors mentioned that a 4hour water treatment experiment is quite long for the stability test of the device, however, previous literature show longer operations (in days) to be performed for long-term stability.

While I am intrigued by the article, I think the above concerns need to be addressed before consideration for publication.

Reviewer #1's Comments:

Reviewer #1: The authors presented 3D-printed mechanical metamaterials for water purification application.

In particular, to achieve robust and high-throughput water purification, the authors designed Douglas fir wood-inspired metamaterial catalysts. With appropriate experimental and simulation data, the concept was successfully demonstrated. Overall the findings look interesting and I would like to suggest several major comments to consider the acceptance of the paper.

1. In the mechanical robustness analysis, analysis of weight was omitted. How light a device can be made should also be considered.

Response to comment: We have normalized the compressive modulus and strength by the relative density which is a data relation with weight in cellular structures, and the results of the specific modulus and specific strength are shown in Fig.2k and Supplementary Information Fig. S12, respectively.

Fig. 2k. Contour map of normalized Young's modulus under different overlap rates and strut diameters for the wood-inspired metamaterial.

Fig. S12. Contour map of normalized strength under different overlap rates and strut diameters for the wood-inspired metamaterial catalyst.

2. Technology review is needed from various perspectives such as weight, strength, price, structure differentiation, and mechanism. Please refer to below papers as prior arts.

“Ultralight Metallic Microlattices”, Science 334, 962, 2011

“ Strong, lightweight, and recoverable three-dimensional ceramic nanolattices”, Science 345, 1322, 2014

“Resilient 3D hierarchical architected metamaterials”, PNAS 112, 11502, 2015

Response to comment: Thank you for the reviewer's suggestions. A summary of the additive manufacturing technology review has been added to the introduction section, with the specific content as follows:

“Thanks to the development of additive manufacturing technology, the manufacturing of highly complex microlattice metamaterials can be realized. However, there are significant differences between different additive manufacturing technologies in terms of material, strength, price, structure, and mechanism. Taking metal additive manufacturing as an example, the lower end of the cost spectrum, such as wire-arc additive manufacturing (WAAM), often has lower performance and accuracy^{1,2}; Medium performance and moderate cost but with good accuracy, such as the selective laser melting (SLM) process using materials such as stainless steel (SS), but requiring

non-conventional particle control settings and limited to manufacturing sizes^{3,4}; Some of the highest-performance metal microlattice production technologies based on two-photon lithography (TPL) and electroplating processes have been fully studied^{5,6}, but are often highly-specialized, time-consuming and cost-intensive. In addition to the 3D printing process, the time required for processes such as polymerization, curing, electroplating, grinding, and etching may exceed 24 hours⁷⁻⁹. To meet the comprehensive requirements of the sewage treatment system for the size, accuracy, strength, transport and catalyst adhesion ability of the support frame, this work adopts a compromise SLM-based 3D printing technology for the manufacturing of 316L stainless steel microlattice metamaterials with different strut diameters and overlap rates.”

3. Please confirm the discrepancy between simulation and experimental results of strain-stress curve (figure 2j).

Response to comment: Figure 2j shows the compression experimental stress-strain results of microlattice metamaterials with 0%, 30%, 50%, and 70% different overlap rates, and no simulation data is available.

4. Please consider and elaborate elastic deformation and failure modes like buckling and fracture. Also, please provide the maximum stress that the structure can withstand.

Response to comment: Firstly, the microlattices that we used belong to the bending-dominated structure, which undergoes buckling deformation when subjected to load; Secondly, the selected material was 316L stainless steel, which has good elastic-plastic properties (Fig. S13). Therefore, our 3D-printed wood-inspired metamaterial structures undergo buckling deformation under the static-quasi-compression process, in which the compressed metamaterial samples are completely squeezed together without any fracture (Fig. S14).

Fig. S13. The stress-strain curves of 316L stainless steel

Fig. S14. Compression deformation behavior of microlattice metamaterials with **a)** 0%, **b)** 30%, **c)** 50%, and **d)** 70% different overlap rates.

We have provided the yield stress of wood-inspired metamaterials with different overlap rates and diameters (Fig. S12). According to the classic Euler buckling formula, the buckling load of a single strut, P_{cr} , is given by the following relationship¹⁰:

$$P_{cr} = \frac{\pi^2 EI}{L^2}$$

where E is the elastic modulus of the matrix material, I is the second moment of inertia and L is the effective length of struts. For the boundary conditions presented in Fig. S15, the critical buckling load for the first buckling mode of struts with circular strut cross-section can be calculated as:

$$P_{cr,UC} = \frac{\pi^2 E d^4}{64 L^4 \sin \theta}$$

Therefore, as the strut diameter, d , increases, and a shorter effective strut length, L , caused by the overlapping process, the yield strength of wood-inspired metamaterials gradually increases.

Fig. S15. Schematic of BCC-based microlattice metamaterials loaded in longitudinal direction. **a)** boundary conditions, the force applied to a quarter model of an arbitrary unit cell and free body diagram of an inclined strut. **b)** The yield strength comparison of microlattice metamaterials with different overlap rates.

We have added the failure mode of the metamaterial structure in the main text, and also have taken the compression process, structural analysis, and other results into the Supplementary Information file.

“Due to the flexibility of the BCC-type microlattice and the plasticity of the 316L material, metamaterial structures all failed as the buckling deformation (Fig. S13-S14).”

“The mechanical analysis of Euler's buckling theory also shows that the decrease in equivalent strut length results in the increase of the buckling force, therefore leading to increased mechanical robustness (Fig. S15).”

5. Please compare the mechanical robustness and transport performance with wood that you referred to. How much can those characteristics be improved when you use metamaterials?

Response to comment: We have added the performance comparison between wood

and wood-inspired metamaterials. Wood and wood-inspired metamaterials have completely different materials and different pore scales. The pore size of Douglas fir (0.1 mm diameter) is about one-seventh that of microlattice metamaterials with 70% overlap rates (0.75 mm diameter) (Fig. S26a). The wood with micrometer-scale pores has good mechanical properties (Fig. S26b-c), but also significantly reduces its transport performance (Fig. S26d). On the contrary, wood-inspired metamaterials with millimeter-scale pores have moderate modulus/strength and excellent transport performances.

Through adopting the overlapping bimodal pore characteristics of this wood, the wood-inspired metamaterial catalyst possessed a wide range of mechanical-transport-catalysis capabilities while a 70% overlap rate has 3X more modulus/strength (Fig. S26e-f) and surface area per unit volume, and 4X normalized reaction kinetics than those of traditional microlattices (Fig. 4a-b).

Micrometer-scale pores in wood make it difficult to conduct permeability tests using the falling head method¹¹. Therefore, we used micro-CT to scan and reconstruct the microstructure of Douglas fir, and used this CT-reconstructed structure for transport simulation to obtain its permeability results. A large number of bimodal pores were also found in the CT-reconstructed microstructure of Douglas fir (Fig. S27). When the flow velocity remained constant, the small pores of the bimodal pores had higher flow velocity and reduced local pressure drop (Fig. S28).

Fig. S26. Comparison between the wood of Douglas fir and wood-inspired metamaterials in a) pore size, b) Young's modulus, c) yield strength and d) apparent permeability. Improvement in e) Young's modulus and f) yield strength through wood-inspired overlapping microlattice design strategy.

Fig. S27. **a-d)** CT-reconstructed wood microstructures: **a)** The CT-reconstructed wood models. **b)** The x-z plane view. **c)** The x-y plane view. **d)** The x-y plane view. **e-g)** The CT-reconstructed region of interest (ROI) models: **e)** the ROI was extracted from the CT-reconstructed wood models. **f)** The x-z plane view. **g)** The x-z plane view. The wood is scanned by the X-ray microscopes (ZEISS Xradia Versa 610). The applied voltage in the micro-CT measurements was 60 kV. These reconstructed models consist of isotropic voxels with $0.5 \mu\text{m}$ and take a scanning period of 150 min. The Air filter was used for scanning the wood sample. These scanning data were further analyzed and visualized using the commercially available image analysis software VG Studio Max 3.4 (Volume Graphics GmbH, Heidelberg, Germany).

Fig. S28. **a-c)** The velocity distribution under different views. **d-f)** the pressure distribution under different views. The upper surface is the inlet boundary and the downward surface is the outlet boundary. Aside from the surfaces of the inlet and outlet, the other walls are set as wall boundaries without slip conditions. The velocity of the inlet-flow side is 0.001 m/s while the outlet pressure of the opposite-flow side is set as zero. The physical properties of fluid with a density of 1000 kg/m^3 and a viscosity of $1.01 \times 10^{-3} \text{ Pa s}$ were assigned to the liquid domain.

6. Figure 3j, please highlight the ultimate goal direction, current state-of-the-art 3D printed structures and the developed metamaterial device.

Response to comment: In Figure 3j, we have distinguished between traditional microlattices (0% overlap rate) and wood-inspired metamaterials (10%-50% overlap rate). Compared to traditional periodic microlattices, our developed wood-inspired metamaterials have higher strength and still good permeability. Besides, overlapping microlattice design enhances the tailoring space for strength and permeability. We also need to note that due to the increase in relative density, the permeability of the

metamaterial structure is inevitably decreased, but for microlattice metamaterials used for structural catalysis, lower permeability means more complete catalytic processes. Due to different application scenarios with different requirements for permeability, we cannot simply provide the ultimate goal direction. For example, the permeability of matter transport in the porous structure of cancellous bone scaffolds mentioned in the introduction only requires permeability values within a limited range of $2.56 \times 10^{-11} \text{ m}^2$ to $74.3 \times 10^{-9} \text{ m}^2$ ¹²⁻¹⁴. Therefore, we did not choose to label the optimization objective of permeability in the graph of permeability versus relative density, but rather present the extensive tailoring ability of the overlap design strategy on permeability. But we also agree with the suggestions of the reviewer, especially from the perspective of the readers. We have also included relevant discussions in the main text as you are aware of the optimization goal of the transport permeability.

High permeability and high relative density are the design goals for robust and high-throughput metamaterials for the applications of high-efficient matter transportation without the reaction between liquid and solid. However, we consider that in the application of metamaterial catalysts, having relatively lower permeability and higher relative density within a certain permeability range is the ultimate goal direction.

Fig. 3j. Comparison of the apparent permeability of microlattice metamaterials under different overlap rates and structural types as a function of relative density.

7. Provide corresponding experimental results for permeability with different shapes of microlattices. (e.g. fluid velocity)

Response to comment: The permeability of the traditional microlattices and different

wood-inspired metamaterials have been experimentally tested (Fig. 3k). The fluid permeability is determined by applying Darcy's law, making sure the Reynolds number (Re) remains sufficiently low (typically $Re < 10$). The adopted permeability setups make use of a falling-head approach, where a tall column of water is placed above the microlattice structure and is allowed to flow through it under the action of gravity¹⁴. The same method has also presented in other literature on permeability testing of porous structures^{11,15-17}.

By using Darcy's law, $v = K\lambda$, the experimental permeability of 3D printed samples can be derived, in which v , K , and λ are the fluid velocity, hydraulic conductivity, and hydraulic gradient, respectively. The initial liquid height H_1 and the final liquid height H_2 were captured at T_0 and T_i , respectively. The hydraulic gradient, λ , is described by:

$$\lambda = \frac{H_1 - H_2}{H}$$

The hydraulic conductivity, K , can be obtained by integrating the water flow rate over the measurement time $T_i - T_0$, as following:

$$K = \frac{aH}{A(T_i - T_0)} \ln\left(\frac{H_1}{H_2}\right)$$

where a and A are the cross-section areas of the standpipe and the samples, respectively.

The experimental permeability of the 3D printed microlattice samples can be inferred by:

$$k_{exp} = \frac{K\mu}{\rho g}$$

The experimental permeability results were compared with the corresponding computational permeability results of the traditional microlattice and different wood-inspired metamaterials, which was presented in Fig. 3k. There is usually a certain deviation in the simulated permeability and experimental permeability, but even if the microlattice topology is different, there is still a high linear relationship under different overlapping designs, indicating a good agreement between the experiment and simulation. The permeability values of CFD simulations were well correlated ($R^2 = 0.80, 0.97, 1.00, 0.94, 0.99, 0.90, \text{ and } 1.00$, for SC, FCC, diamond, cuboctahedron,

octahedron, octet-truss, and BCC, respectively) with experimental results. The discrepancies in experimental and computational permeability values could be attributed to the following reasons: First, the 3D printed microlattices have slight surface roughness, leading to the overestimated permeability. Second, the fluid height could be a factor for the constant pressure permeability testing setup, while the height of the pressure head in the laboratory is lower than the optimal value¹⁷. Finally, the small gap between the permeability testing setup and the periphery of the tested samples also results in permeability discrepancies.

Fig. 3k. Experimental vs. computational permeability results of the traditional microlattices and different wood-inspired metamaterials. The experimental samples for permeability tests are metamaterial catalysts with 0%, 30%, and 50% overlap rates and 0.4 mm strut diameter.

Reviewer #2's Comments:

Reviewer #2: Lei Zhang et al. present a wood-inspired metamaterial catalyst design and structure for water purification. While the main structure is fabricated with 316L stainless steel using metallic 3D printing, the surface area of the structure is doped with cobalt using electrochemical deposition. This approach significantly improves throughput compared to producing the same structure solely using metallic 3D printing. The resulting structures exhibit robustness and efficient permeability for water purification. The paper is well-written and well-presented. The overall utilization of metallic 3D printing and cobalt electrodeposition is a novel approach for water purification. However, there are certain aspects that require further discussion and context in relation to similar technologies.

The paper does not sufficiently elaborate on the advantages and disadvantages of the presented work compared to similar technologies. It is crucial to highlight the advantages to enable researchers to select appropriate methods for different applications. Similarly, it is important to identify the disadvantages to guide future researchers in focusing on areas for technological improvement. While some of these advantages and disadvantages may have been discussed in the authors' prior paper, there is ample opportunity to delve into the trade-offs presented by the metamaterial designs and structures. Furthermore, it would have been beneficial to fabricate and include more complex structures for a comprehensive analysis. I recommend that the authors consider attempting to fabricate such structures or at the very least provide an explanation within the text as to the feasibility of producing more intricate designs (potentially within the conclusions section).

Response to comment: Thank you for the reviewer's suggestions. We have added a review and discussion of similar additive manufacturing technologies, which is shown in the following specific comments.

Then, we have identified the disadvantages to guide future researchers in focusing on areas for technological improvement in the conclusion sections, and specific content as follows:

“Our work focuses on high-throughput structural design to achieve robust and efficient metamaterial catalysts. There are certain limitations in the manufacturing process selection and material selection. However, the design strategy of imitating wood bimodal pores and the strategy of coupling 3D printing with electrochemical deposition to form high-throughput metamaterial catalysts can be used for reference in other flow catalytic fields.”

Below are specific comments, suggestions, and questions that need to be addressed prior to the paper's publication in Nature Communications:

1.a. Considering the innovative approach of creating a wood-based metamaterial structure using stainless 316L through 3D printing, it might be beneficial for the authors to delve further into a comprehensive comparison with analogous technologies that possess the potential for heightened efficiency. Have the authors explored contrasting techniques, including those employing alternative materials such as glass, polymers, and examined their respective advantages and drawbacks? [Patrik Schurch et al., Direct 3D microprinting of highly conductive gold structures via localized electrodeposition, doi.org/10.1016/j.matdes.2023.111780 ; X. Tan et al., Single-Step-Lithography Micro-Stepper Based on Frictional Contact and Chiral Metamaterial, doi.org/10.1002/sml.202202128 ; Grabulosa A. et al, Combining one and two photon polymerization for accelerated high performance (3 + 1)D photonic integration, doi.org/10.1515/nanoph-2021-0733]. Moreover, could a more in-depth exploration of comparable methods shed light on the feasibility and potential advantages of utilizing different materials in achieving similar structural metamaterial outcomes?

Response to comment: Thank you for the reviewer's suggestion. We have added a review of wastewater treatment technologies with 3D printing technologies and different materials in the introduction section. The specific content is as follows:

“Thanks to the development of additive manufacturing technology, the manufacturing of highly complex microlattice metamaterials can be realized. However, there are significant differences between different additive manufacturing technologies

in terms of material, strength, price, structure, and mechanism. Taking metal additive manufacturing as an example, the lower end of the cost spectrum, such as wire-arc additive manufacturing (WAAM), often has lower performance and accuracy^{1,2}; Medium performance and moderate cost but with good accuracy, such as the selective laser melting (SLM) process using materials such as stainless steel (SS), but requiring non-conventional particle control settings and limited to manufacturing sizes^{3,4}; Some of the highest-performance metal microlattice production technologies based on two-photon lithography (TPL) and electroplating processes have been fully studied^{5,6}, but are often highly-specialized, time-consuming and cost-intensive. In addition to the 3D printing process, the time required for processes such as polymerization, curing, electroplating, grinding, and etching may exceed 24 hours⁷⁻⁹. To meet the comprehensive requirements of the sewage treatment system for the size, accuracy, strength, transport and catalyst adhesion ability of the support frame, this work adopts a compromise SLM-based 3D printing technology for the manufacturing of 316L stainless steel microlattice metamaterials with different strut diameters and overlap rates.”

1.b. Given the authors' inclusion of a material comparison table in the supplementary document, which discusses various materials for metamaterial fabrication, could the authors consider integrating relevant insights from this comparison into the main text? This would provide readers with a clearer understanding of the merits of utilizing stainless 316L in their wood-based structure compared to other materials. Additionally, should the authors choose to delve into alternative materials like polymers, could they elucidate the specific attributes that these materials offer and allow readers to identify whether polymers, or metal-organic have the potential to offer performance enhancements for the metamaterial structure [Xueyan Chen et al., Optimal isotropic, reusable truss lattice material with near-zero Poisson's ratio, doi.org/10.1016/j.eml.2020.101048; Furukawa H et al. The chemistry and applications of metal-organic frameworks. Science 2013;341:974–86], particularly when compared

to stainless 316L?

Response to comment: We have used a qualitative radar chart (Fig. 5) to compare the advantages and disadvantages of different materials for wastewater catalysts in terms of manufacturability, structural design, geometry and performance controllability, mechanical strength, and wastewater purification functionality. The supporting literature information for the radar chart is included in the Supplementary Information Table S3, which compares wastewater catalysts of different manufacturing methods and materials. Based on the suggestions of the reviewers, we have added a discussion on manufacturing processes and material types in this section to provide a clearer explanation of the superiority of our SLM-printed Fe/Co-based metamaterials.

“Microlattices or porous structures have strong structural designability and additively manufacturability¹⁸, while also exhibiting significant mechanical robustness for metal-based materials in 3D printing component manufacturing. However, due to the low selectivity of metal-based materials in the SLM process, many materials with good catalytic performance are difficult to manufacture as components, which greatly limits their catalytic applications in the fields of environment and energy. On the contrary, metal-organic frameworks or carbon-based synthetic materials have ultra-high surface area and permanent porosity, making them ideal candidates for fuel (hydrogen and methane) storage, carbon dioxide capture, and catalytic applications¹⁹. However, they are still full of challenges for large-scale engineering application in terms of the current manufacturable sizes of only nanometers/micrometers and poor chemical stability.”

Besides, our wood-inspired microlattice metamaterials are material-independent and primarily optimize mechanical strength, mass transport, and catalytic performance through the structural design strategy. We also think it is certain that high-performance polymers and metal-organic compounds can greatly improve the performance of metamaterial structures, especially the improvement in catalytic efficiency. Our work is to provide a paradigm that enables the decoupling of multiple physical and chemical properties and collaborative regulation through the integration of structure and

function.

2. The authors have conducted a comparison between two distinct structures: The first structure features a traditional microlattice consisting of uniform squares with thin walls composed of inclined struts and robust nodes. In the second structure, a unique approach is taken by superposing microlattices that alter the spatial shape of the internal pore region. Remarkably, the technique of overlapping these microlattices emerges as a promising strategy. This approach, encompassing both large and small pores, exhibits the potential to enhance the overall stiffness of the structure. By reinforcing its ability to withstand the impact of fluid flow, this innovative approach holds significant promise for various applications.

Considering the success of this technique, it could be intriguing to explore simulations, fabrication, or comparative analyses involving structures that employ diverse pore shapes beyond the conventional squares [Huang Y, Lu X, Liang G, Xu Z. Pentamodal property and acoustic band gaps of pentamode metamaterials with different cross-section shapes. *Phys Lett A* 2016;380:1334–8 ; Moein Shafia, Pentamodes : the role of unit cell base topology on mechanical properties, <https://doi.org/10.48550/arXiv.2305.15419>]. Delving into alternative pore geometries, and their corresponding effects on mechanical properties, could provide invaluable insights. For instance, investigating how pore shapes other than squares influence stiffness, resilience, and resistance to fluid flow impact could contribute to an enhanced understanding of how metamaterial structures can be tailored to specific engineering requirements. Such exploration could pave the way for an extended range of applications and further refinement of metamaterial design principles.

Response to comment: Thank you to the reviewers for their recognition of our design strategy for using wood-like overlapping microlattice metamaterials. Regarding the suggestions for considering different pore shapes of microlattice metamaterials and their comparative analysis, Figure 3j has studied the impact of overlapping designs of different topological microlattices on permeability, which possess different pore

shapes. Furthermore, we have added experimental results on the mechanical properties of these microlattice metamaterials as follows:

“At the same time, we should note that for different topological microlattices, the influence of this wood-inspired overlapping design strategy on their mechanical properties is distinctive. In addition to the BCC-type microlattice, we also considered the mechanical responses of other topological microlattices with different pore characteristics (Fig. S16). Overall, the overlapping design strategy can enhance their mechanical robustness. However, not all microlattices follow the rule that the greater the overlap rate, the higher the strength improvement. This is attributed to the limitation of the force acting on the internal strut elements of the microlattice. In addition, it can be found that different topological microlattices can all be used for wood-inspired overlapping design, which verifies the flexibility of our design strategy and the universality of our method, and further enhances the design space of mechanical strength through microlattice topology transformation.”

Fig. S16. The mechanical responses of different traditional topological microlattices and their corresponding wood-inspired metamaterials: **a-f)** stress-strain curve results and **g)** Young's modulus and **h)** yield strength. The traditional microlattices are **a)** diamond, **b)** simple cubic (SC), **c)** octahedron, **d)** face-center cubic (FCC), **e)** octet-truss, and **f)** cuboctahedron, respectively.

3. In the scope of utilizing a stainless steel structure as a substrate for cobalt electrodeposition in water purification applications, have the authors thoroughly investigated the susceptibility of the stainless steel to corrosion phenomena in the intended operational environment? Considering the potential exposure to aqueous solutions and diverse ionic species, have the paper's authors explored how the inherent corrosion resistance of stainless steel could interact with the electrodeposited cobalt

layer, both in terms of synergistic corrosion protection and the possibility of galvanic corrosion [Kong, D. et al., Corrosion of metallic materials fabricated by selective laser melting, doi.org/10.1038/s41529-019-0086-1]? Furthermore, have the authors elucidated how the cobalt deposition process can be harnessed not only for catalytic water purification but also as a means of enhancing the overall corrosion resistance of the stainless steel structure when exposed to the operational conditions of a water treatment setting?

Response to comment: To investigate the effect of cobalt coating on the corrosion resistance of stainless steel (SS)-based metamaterials in the operating environment, the polarization current curve of Co/SS metamaterial catalysts and SS-based metamaterials in 0.2 g/L PMS solution and DI water was tested, as shown in Fig. S24a, and the corrosion potential and corrosion current density were recorded at Fig. S24b-c. It can be seen that in 0.2 g/L PMS solution, the corrosion potential E_c of Co/SS metamaterial catalysts and SS-based metamaterials are -0.58 V and -0.09 V, respectively, where corrosion current I_c of Co/SS metamaterial catalysts and SS-based metamaterials are $0.47 \mu\text{A cm}^{-2}$ and $0.14 \mu\text{A cm}^{-2}$). While in DI water, the corrosion potential E_c of Co/SS metamaterial catalysts and SS-based metamaterials are -0.44 V and -0.13 V, respectively, where corrosion current I_c of Co/SS metamaterial catalysts and SS-based metamaterials are $0.15 \mu\text{A cm}^{-2}$ and $0.035 \mu\text{A cm}^{-2}$). The increased corrosion current in Co/SS is because Co serves as the active species in the PMS activation, thus it is not as resistant to corrosion of PMS as the SS.

According to the standard electrode potential, E^\ominus of Fe/Fe^{2+} is -0.44 V, and E^\ominus of Co/Co^{2+} is -0.28 V, which means that galvanic corrosion occurs when Co/SS (mainly Fe) contact, and electrons can transfer from Fe to Co. This result can hurt the corrosion resistance of SS-based metamaterials as shown in the above corrosion data however, the electron injection into Co is critical to active and circular Co species in the catalytic PMS degradation process (Fig. S24d). This is an important reason for the good catalytic performance of Co/SS metamaterial catalysts. As shown in Fig. S25d, the process of PMS activation will increase the valence state of Co species, and the high-valence state

of Co will inhibit the activation process of PMS to obtain electrons. The galvanic current between Co/SS can inject electrons into Co species to promote the reduction of Co, which accelerates the cycle process of Co species. It is beneficial to improve the activity and prolong the service life. It is worth noting that the SS in metamaterial catalysts is bulky and large, the galvanizing corrosion between SS metamaterials with surface Co will not cause serious damage to the metamaterial catalysts structure.

Fig. S24. **a**) Polarization current curves of Co/SS- and SS-based metamaterial in 0.2g/L PMS solution and DI water, respectively. **b**) Corrosion potential **c**) corrosion current comparison of Co/SS and SS metamaterial in 0.2g/L PMS solution and DI water, respectively. **d**) Schematic diagram of a galvanic current promoting the circulation of Co species.

4. Given the intricate nature of the structure, it's imperative to delve into potential concerns surrounding cracking within this innovative framework. Moreover, the influence of external forces in real-world scenarios prompts an examination of how

these forces could impact the microstructure's stability. In light of these considerations, gaining insights into the following dimensions would undoubtedly enrich the paper's depth:

a. Regarding Potential Cracking: The utilization of stainless steel within the 3D-printed metamaterial structure raises concerns about the propensity for cracks. Has the microstructure been meticulously characterized to identify the presence of existing cracks or anticipate cracks that might manifest during the manufacturing process? Specifically, how has the paper examined the potential influence of incomplete powder fusion during the solidification process on the formation of cracks within the stainless steel microstructure? Moreover, could you elucidate how these considerations are interconnected with the development of the distinctive wood-based metamaterial structure [T. DebRoy et al., Additive manufacturing of metallic components – Process, structure and properties, doi.org/10.1016/j.pmatsci.2017.10.001]?

Response to comment: Thank you for the reviewer's suggestion. We have carefully considered the potential impact of manufacturing defects, such as void defects, cracks, and manufacturing accuracy, in the SLM process on the performance of metamaterial catalysts.

1) For void defects, we conducted micro-CT void defect analysis on the metamaterial structures manufactured by SLM with different overlap rates (Fig. S5). Firstly, we analyzed the void defects of overall CT-reconstructed metamaterial models. We found that the stainless steel-based metamaterial structures manufactured by SLM have extremely low void defects. For the $15 \times 15 \times 15 \text{ mm}^3$ metamaterial structure with a void defect volume controlled at below $5 \times 10^{-4} \text{ mm}^3$, the peak void volume ranges from 2.5 to $3.5 \times 10^{-5} \text{ mm}^3$. As the overlap rate of the metamaterial increases, the volume of the void defects decreases, which may be due to the tight microlattice structure arrangement, which reduces the accumulation of heat in different regions of the metamaterials, thereby reducing the void defects. 316L stainless steel is a mature raw material for the SLM process. Then, we also used a strut element in the microlattice metamaterials as the region of interest (ROI) for local pore defect analysis (Fig. S6).

The volume of voids is also maintained at a relatively low level, below 10^{-3} mm³, and as the degree of metamaterial overlap increases, the void defects tend to decrease. The distribution of void defects is mainly located in the central area of the strut element. In addition, suspended molten or semi-molten metal powder particles can also be observed. This is a limitation of the manufacturing accuracy of SLM-printed microlattice metamaterials.

From the results of the CT analysis, it can be seen that the void defects of wood-inspired metamaterials made of 316L stainless steel are controlled at an extremely low level and do not affect the mechanical properties of the metamaterials. Therefore, we consider that the void defect is almost negligible.

2) For crack defects, we observed the surface morphology of microlattice metamaterials manufactured by SLM using scanning electron microscopy and did not find any crack defects (Fig. S7). This is attributed to the excellent additive manufacturing processability of 316L stainless steel substrate.

3) For manufacturing accuracy, we have compared and analyzed the surface morphology and surface deviation in two-dimensional slices of SLM-built microlattice metamaterials (Fig. S8). The original micro-CT sections indicate that there is a significant phenomenon of metal powder adhesion on the downward side of the strut elements in the microlattice metamaterials. Two-dimensional surface deviation contours of the microlattices based on the CT data and original models are presented in Fig. S8e-h. The thin color lines perpendicular to the surfaces of isolated struts signify the deviations while the length and the color type of these lines signify the magnitude of the manufacturing surface deviation. It can be observed that the adhesive particles on the lower surface are the main cause of manufacturing deviation. Fig. S8i-l shows the statistical surface deviations and their distributions for the traditional microlattices and wood-inspired microlattice metamaterials with different overlap rates. The surface deviations of all microlattices were Gaussian distributed. The peak deviations of the microlattices with 0%, 30%, 50%, and 70% overlap rates were 14 μ m, 17 μ m, 23 μ m, and 25 μ m, respectively. The D-values, D10, D50, and D90, i.e., the intercepts for 10%,

50%, and 90% of the cumulative percentages based on the surface area of the traditional microlattices, were -13 μm , 29 μm , and 93 μm , respectively. The corresponding values for the wood-inspired microlattice metamaterials with 30% overlap rates were -19 μm , 30 μm , and 102 μm . The corresponding values for the wood-inspired microlattice metamaterials with 50% overlap rates were -23 μm , 35 μm , and 119 μm . The corresponding values for the wood-inspired microlattice metamaterials with 30% overlap rates were -27 μm , 37 μm , and 129 μm . It can be found that whether it is the peak deviation or the cumulative deviation, as the overlap rate increases, the absolute value of the deviation shows a gradually increasing trend. In general, wood-inspired microlattice metamaterials with larger overlap rates exhibit a longer thermal history on the interaction between the laser and powder²⁰. The alternating heating and cooling processes led to more notable adhesion of the powder to the struts²¹. This adhesion was probably the reason for the higher positive deviation in the wood-inspired metamaterials than in traditional microlattices.

However, when using SLM-manufactured microlattice metamaterials for catalytic applications, the manufacturing accuracy would increase the available surface area of metamaterial catalysts (Fig. S9), thereby improving catalytic efficiency. It can be observed that as the overlap rate increases, the improvement rate of surface area of SLM-manufactured metamaterials also gradually increases (Fig. S9). Correspondingly, the normalized degradation kinetic constant increases with the increase in overlap rate (Fig. 4a).

Fig. S5. The void defect distribution in the **a**) traditional microlattice with 0% overlap rate and wood-inspired metamaterials with **b**) 30%, **c**) 50%, and **d**) 70% overlap rates. **e**) The comparison of void volume versus counts curves between the traditional microlattice and the wood-inspired metamaterials.

Fig. S6. The void defect analysis of the region of interest (ROI). **a**) The ROI Schematic model. **b**) The distribution of void defect volume versus their corresponding diameters. **c**) the raw CT-constructed ROI models and **d**) void defect locations for the microlattice (**ci** and **di**) and wood-inspired metamaterials (**cii-civ** and **dii-div**).

Fig. S7. The surface morphologies of the **a**) traditional microlattice with 0% overlap

rate and wood-inspired metamaterials with **b)** 30%, **c)** 50% and **d)** 70% overlap rates.

Fig. S8. **a-d)** The CT-reconstructed 2D slicing model, **e-h)** surface deviation analysis, and **i-l)** the manufacturing deviation data of the **a)** traditional microlattice with 0% overlap rate and wood-inspired metamaterials with **b)** 30%, **c)** 50% and **d)** 70% overlap rates.

Fig. S9. Surface area comparison between CAD models and CT-reconstructed models for the microlattices and wood-inspired metamaterials with different overlap rates.

b. Addressing External Load Effects: The operational reality of the metamaterial involves encountering external forces, notably water flux in the context of water purification. How has the paper accounted for the impact of such external loads on the structural integrity and stability of the stainless steel microstructure? Additionally, considering the significance of the chosen overlap rates, I'm intrigued to understand whether varying external load conditions could give rise to different modes of crack separation. Can you shed light on the interplay between the microstructure's mechanical response, the influence of water flux, and the potential diversification of crack propagation mechanisms, contingent on the selected overlap rates [N. Shamsaei, A. Yadollahi, L. Bian, S.M. Thompson, *Addit. Manufac.* 8 (2015): p. 12-35, doi.org/10.1016/j.addma.2015.07.002]?

Response to comment: As stated in the previous reply, there are no crack defects in our metamaterial structure, but we agree with the reviewer's consideration of the impact of SLM manufacturing defects on structural integrity and stability under external loads. We have performed mechanical simulation compression on the CT-reconstructed models of the raw SLM-manufactured samples, and compared them with the CAD structural model, with one end fixed and the other end subjected to a pressure of 1 Pa (Fig. S10a). We found that the stress level of SLM-manufactured structures is greater than that of CAD model structures (Fig. S10b), which could be attributed to the local surface deviations and void defects. For both CAD models and CT-reconstructed models, the increasing overlap rate of microlattices gradually increases their structural stiffness, resulting in an increase in the maximum stress inside the structure. Besides, the slight manufacturing deviations and negligible pore defects do not significantly alter the stress distribution, but only marginally increase the degree of stress concentration (Fig. S10c-f).

The internal relationship among the manufacturing deviation, transport performance, and mechanical properties is described as follows: Since there are no crack defects and only a small amount of internal void defects in this work, the impact on the fluid transport process of microlattice metamaterials can be neglected. The

manufacturing deviations produced during the SLM process are mostly positive dimensional deviations, which means that the flow velocity inside the microlattice would locally increase, and the complexity of the flow field would also increase, resulting in a decrease in permeability; however, positive deviations would thicken the volume of the struts, thereby enhancing its mechanical robustness against impact. Therefore, the SLM manufacturing deviation would slightly improve mechanical performance and slightly reduce the transport performance of wood-inspired microlattice metamaterials.

Figure S10. Von Mises distribution comparison between CAD models and CT-reconstructed model for traditional microlattices and wood-inspired metamaterials. **a)**

Boundary condition. **b)** Maximum Von mises stress comparison of wood-inspired metamaterials under different overlap rates and traditional microlattices. Stress distribution of **c)** the CAD model and **d)** the CT-reconstructed model of the traditional microlattice with 0% overlap rate. Stress distribution of **e)** the CAD model and **f)** the CT-reconstructed model of the wood-inspired metamaterial with 50% overlap rate.

5. Minor Comments

Page 3, line 57, "transportataion" -> "transportation"

Page 4, line 97, "achievinge" -> "achieving"

Response to comment: Thank you for the reviewer's reminder. The spelling errors have been corrected.

Reviewer #3's Comments:

Reviewer #3: This manuscript reports a new way of creating water purification devices using metamaterials design. The structural configurations with highly stable material properties generated by the wood-inspired design can potentially be important.

Positively, this is highly relevant and highly unique to the current water treatment practices. It will really have an impact on how we manufacture water treatment devices. The bottom up approach of designing the device using 3D printing and including catalysts in them will significantly enable the new industrial manufacturing of water filtration devices.

1. The major concern is that why the authors are calling the 3D printed structure a metamaterial while all the other papers mentioned in the manuscript (showing comparative) results didn't although all of these recent papers and this manuscript are using similar 3D printing approach.

Response to comment: Metamaterials refer to artificial structures or composite materials with extraordinary physical properties that cannot be found in natural materials, such as electromagnetic metamaterials, mechanical metamaterials, acoustic metamaterials, etc. Recently, the concept of metamaterials has been further expanded and applied to more fields, such as flexible metamaterial electronics²², robotic metamaterials²³, etc. Our work is an attempt at a combination of metamaterial structural design and catalyst design in the field of environment.

Traditional lattice structures are classified according to their basic units, including strut-based lattice structures^{24,25}, plate-based lattice structures^{26,27}, and minimal surface-based lattice structures^{28,29}. For traditional lattice structures, the regulation of their different physical properties is often coupled and contradictory, making it difficult to achieve wide domain regulation^{30,31}. Through analytical, numerical, and experimental verification, we found that our designed 3D wood-inspired overlapping microlattice structures exhibit decoupled mechanical-transport-catalytic performances that cannot be achieved in a single standard traditional lattice structure, and achieve wide tailoring ranges of mechanical, transport, and catalytic performances, including modulus,

strength, permeability, and catalytic efficiency. Therefore, we refer to this structure with robust and high-throughput properties as a metamaterial catalyst.

2. The authors suggested that the metamaterial structure help to slow down the water permeability. I am not sure if that is a desirable output for high volume water industry. If the authors can justify the application space it would be great.

Response to comment: The design strategy of wood-inspired overlapping microlattices would deepen the interlacing degree of the struts inside the metamaterial structure, increase the tortuosity of fluid penetration, and thus reduce the flow rate to a certain extent. However, an increase in the tortuosity of fluid penetration would significantly increase the contact reaction time between the fluid and the metamaterial catalyst. For traditional microlattice structures with large pores and fast flow rates, the catalytic reaction time is limited. Our catalytic experiments also showed that the 70% overlap metamaterial catalyst showed the highest reaction kinetic constant per unit volume after normalization (Fig. 4a-b). This work utilizes wood-inspired overlapping design strategies to regulate the synergistic mechanical–transport-catalytic relationship.

Different pipeline sizes would have different fluid transport fluxes at the same flow rate, which would result in different mechanical-transport-catalytic regulation requirements. We have also considered this point, and therefore, in Figure 5g, the possibility of the widespread application of this design concept under different scale and robustness application conditions is illustrated. This work provides a paradigm for mechanical-transport-catalytic multi-performance regulation through a wood-inspired bimodal structure, providing a design strategy for flow catalysis applications in different scenarios that can be used for reference.

3. The authors mentioned that a 4 hour water treatment experiment is quite long for the stability test of the device, however, previous literature show longer operations (in days) to be performed for long-term stability.

Response to comment: We thank the reviewer for acknowledging that our water

treatment is already quite long. To explore the maximum stability of Co/SS metamaterial catalysts, a water treatment system as shown in Fig. S26 was assembled, which consisted of two peristaltic pumps mixing 20 ppm and 0.4 g/L PMS at a flow rate of 0.5 mL/min respectively into a fixed bed processor equipped with Co/SS metamaterial catalysts, and a third bucket was used to collect the treated liquid. It was found that the Co/SS metamaterial catalysts could run continuously for more than 96 h and the degradation efficiency remained above 90%, as shown in Fig. S26b. After 108 h treatment, the degradation efficiency decreased, which may be due to the occurrence of surface deactivation and passivation. This deactivation is attributed to the lack of active Co species, and the degradation efficiency can be restored to 98% by Ar/H₂ reduction at 300°C for 3h. It should be noted that a vast of AOP-activated materials have only studied the service life of 5 cycles, and this stability is far from enough for practical water treatment materials. Our Co/SS metamaterial catalysts show the stability of the day level, which has a certain positive significance for the practical application of water treatment materials. Compared with other high-stability water treatment systems, the stability of our system has a considerable advantage (Table S2).

Fig. 4 **j)** Schematic diagram of long-time water treatment device. **k)** Long-term performance of Co/SS metamaterial catalyst for SMX degradation.

Table S2. Summarizes the stability working time for some catalysts

No.	Catalyst	Pollutant	Pollutant concentration (mg/L)	Oxidant concentration	Reaction time	Ref.
1	Co/SS metamaterial catalysts	SMX	10	PMS 0.2 g/L	4 d	This work
2	zeolite@ZIF-67 composites	TC	50	PMS 0.5 g/L	300 min	10
2	3DP MG/30Cu	RHB	20	H ₂ O ₂ 2 mM	over 100 cycles	22
3	ce-MoS ₂	BPA	2	PMS 0.05 g/L	360 min	23
4	CoS@FeS-1	SMX	10	PMS 1 mM	15 h	24
5	Co-TPML	BPA	50 μM	PMS 20 mM	60 min	25
6	BA/MoS ₂ @CS H	DC	40	PMS 0.333 g/L	3500 min	26

References

- 1 Ho, A. *et al.* On the origin of microstructural banding in Ti-6Al4V wire-arc based high deposition rate additive manufacturing. *Acta Materialia* **166**, 306-323, doi:10.1016/j.actamat.2018.12.038 (2019).
- 2 Liu, G. *et al.* Additive manufacturing of structural materials. *Materials Science and Engineering: R: Reports* **145**, doi:10.1016/j.mser.2020.100596 (2021).
- 3 Ren, J. *et al.* Strong yet ductile nanolamellar high-entropy alloys by additive manufacturing. *Nature* **608**, 62-68, doi:10.1038/s41586-022-04914-8 (2022).
- 4 Olakanmi, E. O., Cochrane, R. F. & Dalgarno, K. W. A review on selective laser sintering/melting (SLS/SLM) of aluminium alloy powders: Processing, microstructure, and properties. *Progress in Materials Science* **74**, 401-477, doi:10.1016/j.pmatsci.2015.03.002 (2015).
- 5 Tan, X. *et al.* Single-Step-Lithography Micro-Stepper Based on Frictional Contact and Chiral Metamaterial. *Small* **18**, e2202128, doi:10.1002/sml.202202128 (2022).
- 6 Grabulosa, A., Moughames, J., Porte, X. & Brunner, D. Combining one and two photon polymerization for accelerated high performance (3 + 1)D photonic integration. *Nanophotonics* **11**, 1591-1601, doi:10.1515/nanoph-2021-0733 (2022).
- 7 Schaedler, T. A. *et al.* Ultralight Metallic Microlattices. *Science* **334**, 962-965 (2011).
- 8 Meza, L. R., Das, S. & Greer, J. R. Strong, lightweight and recoverable three—dimensional ceramic nanolattices. *Science* **345**, 1322-1326 (2014).
- 9 Meza, L. R. *et al.* Resilient 3D hierarchical architected metamaterials. *Proc Natl Acad Sci U S A* **112**, 11502-11507, doi:10.1073/pnas.1509120112 (2015).
- 10 Ahmadi, S. M. *et al.* Mechanical behavior of regular open-cell porous biomaterials made of diamond lattice unit cells. *Journal of the Mechanical Behavior of Biomedical Materials* **34**, 106-115, doi:10.1016/j.jmbbm.2014.02.003 (2014).
- 11 Zhang, L., Song, B., Yang, L. & Shi, Y. Tailored mechanical response and mass transport characteristic of selective laser melted porous metallic biomaterials for bone scaffolds. *Acta Biomaterialia* **112**, 298–315, doi:10.1016/j.actbio.2020.05.038 (2020).
- 12 Morgan, E. F. & Keaveny, T. M. Dependence of yield strain of human trabecular bone on anatomic site. *Journal of Biomechanics* **34**, 569-577 (2001).
- 13 Linde, F. & Hvid, I. The effect of constraint on the mechanical behaviour of trabecular bone specimens. *Journal of Biomechanics* **22**, 485-490 (1989).
- 14 Bobbert, F. S. L. *et al.* Additively manufactured metallic porous biomaterials based on minimal surfaces: A unique combination of topological, mechanical, and mass transport properties. *Acta Biomaterialia* **53**, 572-584, doi:10.1016/j.actbio.2017.02.024 (2017).
- 15 Li, Y. *et al.* Additively manufactured functionally graded biodegradable porous iron. *Acta Biomaterialia* **96**, 646-661, doi:10.1016/j.actbio.2019.07.013 (2019).
- 16 Zhang, X. Y., Fang, G., LeeFlang, S., Zadpoor, A. A. & Zhou, J. Topological design, permeability and mechanical behavior of additively manufactured functionally graded porous metallic biomaterials. *Acta Biomaterialia* **84**, 437-452, doi:10.1016/j.actbio.2018.12.013 (2019).
- 17 Montazerian, H., Zhianmanesh, M., Davoodi, E., Milani, A. S. & Hoorfar, M. Longitudinal and radial permeability analysis of additively manufactured porous scaffolds: Effect of pore shape and porosity. *Materials & Design* **122**, 146-156, doi:10.1016/j.matdes.2017.03.006 (2017).
- 18 Chen, X. *et al.* Optimal isotropic, reusable truss lattice material with near-zero Poisson's ratio.

- Extreme Mechanics Letters* **41**, doi:10.1016/j.eml.2020.101048 (2020).
- 19 Furukawa, H., Cordova, K. E., O'Keeffe, M. & Yaghi, O. M. The chemistry and applications of
metal-organic frameworks. *Science* **341**, 1230444, doi:10.1126/science.1230444 (2013).
- 20 Yang, L. *et al.* An investigation into the effect of gradients on the manufacturing fidelity of triply
periodic minimal surface structures with graded density fabricated by selective laser melting.
Journal of Materials Processing Technology **275**, 116367,
doi:10.1016/j.jmatprotec.2019.116367 (2020).
- 21 Al-Ketan, O., Rowshan, R. & Abu Al-Rub, R. K. Topology-mechanical property relationship of
3D printed strut, skeletal, and sheet based periodic metallic cellular materials. *Additive
Manufacturing* **19**, 167-183, doi:10.1016/j.addma.2017.12.006 (2018).
- 22 Jiang, S. *et al.* Flexible Metamaterial Electronics. *Adv Mater* **34**, e2200070,
doi:10.1002/adma.202200070 (2022).
- 23 Cui, H. *et al.* Design and printing of proprioceptive three-dimensional architected robotic
metamaterials. *Science* **376**, 1287-1293 (2022).
- 24 Maconachie, T. *et al.* SLM lattice structures: Properties, performance, applications and
challenges. *Materials & Design* **183**, 108137, doi:10.1016/j.matdes.2019.108137 (2019).
- 25 Tancogne-Dejean, T. & Mohr, D. Elastically-isotropic truss lattice materials of reduced plastic
anisotropy. *International Journal of Solids and Structures* **138**, 24-39,
doi:10.1016/j.ijsolstr.2017.12.025 (2018).
- 26 Tancogne-Dejean, T., Diamantopoulou, M., Gorji, M. B., Bonatti, C. & Mohr, D. 3D Plate-
Lattices: An emerging class of low-density metamaterial exhibiting optimal isotropic stiffness.
Advanced Materials **30**, 1803334, doi:10.1002/adma.201803334 (2018).
- 27 Duan, S., Wen, W. & Fang, D. Additively-manufactured anisotropic and isotropic 3D plate-
lattice materials for enhanced mechanical performance: Simulations & experiments. *Acta
Materialia* **199**, 397-412, doi:10.1016/j.actamat.2020.08.063 (2020).
- 28 Feng, J., Fu, J., Yao, X. & He, Y. Triply periodic minimal surface (TPMS) porous structures:
from multi-scale design, precise additive manufacturing to multidisciplinary applications.
International Journal of Extreme Manufacturing **4**, doi:10.1088/2631-7990/ac5be6 (2022).
- 29 Xu, Y., Pan, H., Wang, R., Du, Q. & Lu, L. New families of triply periodic minimal surface-like
shell lattices. *Additive Manufacturing* **77**, doi:10.1016/j.addma.2023.103779 (2023).
- 30 Callens, S. J. P., Arns, C. H., Kuliesh, A. & Zadpoor, A. A. Decoupling minimal surface
metamaterial properties through multi-material hyperbolic tilings. *Advanced Functional
Materials* **31**, 2101373, doi:10.1002/adfm.202101373 (2021).
- 31 Zhang, L. *et al.* Decoupling Microlattice Metamaterial Properties Through a Structural Design
Strategy Inspired by the Hall–Petch Relation. *Acta Materialia* **238**, 118214,
doi:10.1016/j.actamat.2022.118214 (2022).

REVIEWERS' COMMENTS

Reviewer #1 (Remarks to the Author):

I appreciate authors' effort to improve the manuscript quality upon the reviewers' comments. I believe that the improved figures and text justify the publication of this paper in Nature Communications.

Reviewer #3 (Remarks to the Author):

I looked through the revisions and very happy with the revisions by the authors. I recommend publishing this manuscript.

Reviewer #1's comments:

1. I appreciate authors' effort to improve the manuscript quality upon the reviewers' comments. I believe that the improved figures and text justify the publication of this paper in Nature Communications.

Answer: Thank you for providing the modification suggestions.

Reviewer #3's comments:

1. I looked through the revisions and very happy with the revisions by the authors. I recommend publishing this manuscript.

Answer: Thanks for the review's comments.